# XRPO: Pushing the limits of GRPO with Targeted Exploration and Exploitation

**Udbhav Bamba** [* 1]  **Minghao Fang** [* 2]  **Yifan Yu** [* 2]  **Haizhong Zheng** [3]  **Fan Lai** [2]

## Abstract

Reinforcement learning algorithms such as GRPO have driven recent advances in large language model (LLM) reasoning. While scaling the number of rollouts stabilizes training, existing approaches suffer from limited exploration on challenging prompts and leave informative feedback signals underexploited, due to context-independent rollout allocation across prompts (e.g., generating 16 rollouts per prompt) and relying heavily on sparse rewards. This paper presents *XRPO* (eXplore–eXploit GRPO), a framework that recasts policy optimization through the principled lens of rollout exploration–exploitation. To enhance exploration, *XRPO* introduces a mathematically grounded rollout allocator that adaptively prioritizes prompts with higher potential for uncertainty reduction. It further addresses stagnation on zero-reward prompts through an in-context seeding strategy that injects curated exemplars, steering the model into more difficult reasoning trajectories. To strengthen exploitation, *XRPO* develops a group-relative, novelty-aware advantage sharpening mechanism that leverages sequence likelihoods to amplify low-probability yet correct responses. Experiments across diverse math and coding benchmarks demonstrate that *XRPO* outperforms existing advances (e.g., GRPO and GSPO) up to 4% pass@1 and 6% cons@32, while accelerating training convergence by $2.7\times$. Our code and datasets are publicly available at https://github.com/UIUC-MLSys/XRPO.

## 1. Introduction

Recent breakthroughs in applying reinforcement learning (RL) to large language models, such as GPT-o3 (OpenAI,

2025), Qwen3 (Yang et al., 2025a), and Deepseek-R1 (Guo et al., 2025), have demonstrated its effectiveness in enhancing reasoning capabilities. A key driver of this progress has been reinforcement learning with verifiable rewards (RLVR), where models receive rule-based numerical feedback on their generations. RLVR, exemplified by GRPO (Shao et al., 2024) and its very recent extensions (e.g., GSPO (Zheng et al., 2025a)), has emerged as a primary pathway.

Despite rapid progress, RLVR continues to face persistent challenges—most notably slow training and sparse feedback—that form fundamental bottlenecks to both efficiency and quality. We categorize these challenges through the lens of exploration and exploitation: expanding exploration into uncertain rollout regions, and maximizing exploitation of known informative behaviors:

- *Under-exploration of valuable rollouts in generation.* Existing methods (e.g., GRPO and GSPO) use static rollout allocation across prompts (e.g., generating 16 rollouts per prompt), which dilutes valuable signals from high-reward-variance rollouts and leaves zero-accuracy prompts underexplored, whose mastery is critical for surpassing current performance limits. Recent dynamic sampling approaches attempt to gather learning signals by over-sampling or discarding prompts with accuracies of 1 or 0 (Yu et al., 2025a). However, despite large computational overhead (e.g., generating multiple rollouts and only retaining a few (Hou et al., 2025)), these methods lack differentiated exploration and ignore how prompts vary in their potential to expand the model's decision boundary. Moreover, discarding zero-accuracy prompts, often hard questions beyond the model's current capability, risks the model never pushing past limits.

- *Under-exploitation of trajectory signals in rewards.* Simple rule-based rewards (e.g., 1 for correct response, otherwise 0) collapse distinctions among rollouts. Yet, the rich information embedded in generation trajectories remains largely underexploited, which suppresses the model's ability to explore the broader decision space effectively. Recent advances have attempted to enrich exploration via step-wise dense rewards and tree sampling (Hou et al., 2025; Yang et al., 2025b), but these approaches incur high overhead (e.g., sampling many additional rollouts) and still rely on coarse heuristics to

---

[*]Equal contribution  [1]Unaffiliated  [2]University of Illinois Urbana–Champaign  [3]Carnegie Mellon University. Correspondence to: Fan Lai <fanlai@illinois.edu>.

*Proceedings of the 43rd International Conference on Machine Learning*, Seoul, South Korea. PMLR 306, 2026. Copyright 2026 by the author(s).

model the sampling space.

To address these limitations, we propose *XRPO* (eXplore–eXploit GRPO), a novel rollout optimization framework that systematically balances exploration and exploitation in RLVR. By prioritizing informative rollout exploration while sharpening exploitation signals from successful trajectories, *XRPO* breaks through the edge of model capability and enables more stable and faster RLHF convergence. Our key contributions are threefold:

- *Novel Hierarchical Rollout Exploration*: We introduce a hierarchical rollout planner that adaptively allocates rollout budgets based on uncertainty reduction and exploration bonuses. This design focuses high-reward variance prompts near the decision boundary, where additional rollouts are most informative. To address degenerate prompt groups where all responses fail and gradients vanish, we further seed these hard prompts with curated in-context exemplars drawn from an evolving corpus of verified successful rollouts. These mechanisms ensure that both ambiguous and previously unsolved prompts contribute meaningful learning signals, breaking symmetry and expanding the capability frontier.

- *Novelty-Guided Advantage Sharpening*: To improve exploitation, *XRPO* introduces a sequence-level novelty measure that augments standard advantage estimation. Rollouts that are correct yet atypical under the model's own distribution receive an entropy-inspired bonus, enabling the learner to distinguish among superficially similar successes. This shaping promotes generalization to underexplored reasoning paths and counteracts the homogenization induced by sparse, rule-based rewards.

- *Comprehensive Evaluation*: We conduct extensive experiments on challenging math reasoning and code generation benchmarks. *XRPO* consistently outperforms vanilla GRPO and recent advances (e.g., DAPO and GSPO), achieving 4% absolute gains in pass@1 accuracy and $2.4\times$ faster training convergence.

## 2. Background and Related Works

### 2.1. Group Relative Policy Optimization

Recent advances in RLHF have introduced Reinforcement Learning with Verifiable Rewards (RLVR), which typically relies on sparse, rule-based numerical feedback and therefore requires generating many rollouts per prompt to estimate trajectory-level advantages. Within this paradigm, Group Relative Policy Optimization (GRPO) (Shao et al., 2024) has emerged as a dominant approach. It incorporates rule-based reward functions and group-relative advantage:

$$\mathcal{J}(\theta) = \mathbb{E}_{(q,\mathbf{o})\sim\mathcal{D}_{\theta_{\text{old}}}} \left[ \frac{1}{G} \sum_{i=1}^{G} \min \left( \rho_i(\theta) A_i, \right. \right.$$

$$\left. \left. \text{clip} \left( \rho_i(\theta), 1 - \epsilon, 1 + \epsilon \right) A_i \right) - \beta D_{\text{KL}}(\pi_\theta \| \pi_{\text{ref}}) \right]$$

where $\rho_i = \frac{\pi_\theta(o_i|q)}{\pi_{\theta_{\text{old}}}(o_i|q)}$ is the importance ratio and $A_i = \frac{R(q,o_i)-\mu_R}{\sigma_R}$ is the group-normalized advantage, with $\mu_R$ and $\sigma_R$ being the mean and standard deviation of rewards $\{R(q,o_j)\}_{j=1}^{G}$.

Here, $\mathbf{o} = \{o_1, \ldots, o_G\}$ represents $G$ response rollouts sampled from $\pi_{\theta_{\text{old}}}(\cdot \mid q)$ for each prompt $q$. This formulation naturally handles reward scaling but encounters difficulties when all responses yield identical rewards (zero variance), resulting in undefined advantages that provide no training signal, a critical challenge for optimization.

### 2.2. Related Works

**Extending GRPO with Improved Optimization.** Several methods extend GRPO to address its optimization limitations. GSPO (Zheng et al., 2025b) refines the importance ratio computation using sequence likelihood and performs sequence-level clipping, yet still under-explores rollouts critical to expanding the model's capability frontier. TreePO (Li et al., 2025c) and TreeRL (Hou et al., 2025) reformulate GRPO's generation process as tree-structured search, providing stepwise supervision through branching policies. However, these tree-based extensions introduce substantial overhead by generating many additional rollouts. In contrast, *XRPO* improves GRPO by dynamically allocating rollout resources via a mathematically grounded allocator and shaping advantages based on trajectory novelty, achieving better exploration without excessive computational cost.

**Dynamic Sampling Strategies for GRPO.** The sparse reward structure in GRPO has motivated specialized sampling strategies to improve gradient efficiency (Fatemi et al., 2025; Li et al., 2025b; Wang et al., 2025c). DAPO (Yu et al., 2025a) introduces dynamic sampling that over-samples informative prompts while filtering out those with fully correct or fully incorrect rollouts. GRESO (Zheng et al., 2025c) leverages prior reward dynamics to skip uninformative prompts entirely. Beyond these representative methods, related work has also explored adaptive data allocation and filtering strategies from multiple perspectives, such as difficulty-based allocation in CurES (Zeng et al., 2026) and E3-RL4LLMs (Liao et al., 2025), gradient-variance allocation in GVM-RAFT (Yao et al., 2026), offline data curation in DART-Math (Tong et al., 2024), prompt-level selection in MoPPS (Qu et al., 2026) and Knapsack RL (Li et al., 2025d). These methods primarily operate at the prompt selection or

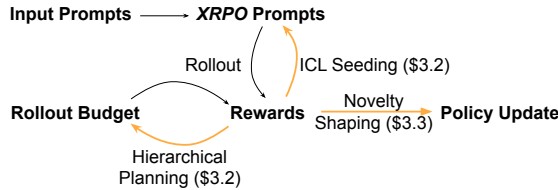

*Figure 1.* Overview of the *XRPO* framework.

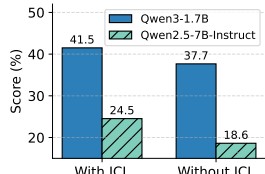

*(a)* Accuracy gains.

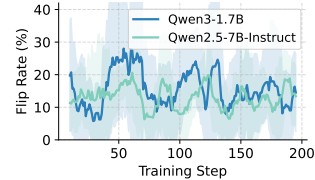

*(b)* Reduces zero-reward groups.

*Figure 2.* **(a)** ICL improves performance on the DAPO training dataset. **(b)** ICL recovers a significant portion of previously unsolved prompts, improving generalization to harder problems.

static allocation level. *XRPO* complements these approaches by operating at the rollout allocation level: given a batch of prompts, it dynamically distributes rollout budgets based on runtime uncertainty estimates, generating more valuable trajectories for both exploration and learning.

**Self-Refinement and In-Context Learning.** Beyond optimization improvements, several works explore how LLMs can leverage feedback to refine their outputs. Natural Language Feedback approaches (Hancock et al., 2019; Wang et al., 2025d; Chen et al., 2024) use human annotators or stronger models to guide refinement, though at significant cost. Multi-turn RL methods (Chen et al., 2023; Kumar et al., 2024; Huang et al., 2023) prompt models to iteratively review their own responses. In multi-turn agent settings, CEL (Wang et al., 2025a) induces environment rules from interaction episodes, and LaMer (Jiang et al., 2025) learns exploration strategies via meta-RL with in-context adaptation. RePO (Li et al., 2025a) uses experience replay buffers of past trajectories. These methods target different settings (multi-turn or replay-based) and are orthogonal to our single-turn RLVR pipeline. In-Context Learning (ICL) (Yu et al., 2026; 2025b) offers an alternative by conditioning generation on exemplars without parameter updates. We adopt ICL seeding to break zero-reward symmetry on hard prompts by injecting successful rollouts from an evolving corpus as contextual guidance. To our knowledge, this is the first demonstration of ICL's effectiveness within RLVR training.

## 3. *XRPO* Method

This paper introduces *XRPO*, as shown in Figure 1, which extends GRPO's capabilities from an exploration–exploitation perspective. We next start by identifying specific limitations in each dimension that motivate our work.

### 3.1. Opportunities for Better Exploration-Exploitation

Existing GRPO-based methods often adopt a static strategy for both rollout allocation and reward assignment, uniformly distributing rollout budgets across prompts and assigning sparse, rule-based rewards across rollouts. This rigid approach creates two key inefficiencies: insufficient exploration of valuable prompts and inadequate exploitation of

the trajectories of generated answers.

**Exploring uncertain, edge-of-policy prompts.** Uniformly allocating generation budgets across prompts suppresses the influence of edge-case prompts with high reward variance, which often yield the steepest advantage gradients. At the same time, complex prompts that consistently receive zero rewards become persistent learning bottlenecks: they produce no gradient signal, yet mastering them is essential for expanding the model's capability frontier. To break this symmetry and enable effective exploration on such hard prompts, we can explore leveraging In-Context Learning (ICL) as a seeding mechanism that temporarily expands the search space. By injecting curated exemplars, ICL uncovers reasoning strategies that are otherwise inaccessible under sparse rewards, particularly when correct trajectories are rare but critical for guiding refinement. As shown in Figure 2a, ICL improves accuracy on Qwen3-1.7B from 37.7% to 41.5%, while Figure 2b shows that it corrects 15–20% of previously zero-accuracy prompts. These results demonstrate that ICL can substantially enlarge the exploration space and improve generalization to harder problems.

However, applying ICL to GRPO introduces new challenges: identifying which prompts genuinely benefit from seeding, selecting exemplars that expand rather than bias the search space, and integrating ICL without incurring excessive overhead or destabilizing training. These challenges require a principled, selective ICL seeding strategy tightly coupled with rollout allocation.

**Exploiting trajectory differences in rewards.** Rule-based rewards in GRPO assign identical advantage to all correct/wrong rollouts, regardless of the underlying reasoning strategy, leading to sparse or even degenerate learning signals. We posit that correct solutions with inherent, relatively low likelihood under the model's current distribution offer the strongest learning signal for LLM reasoning. By emphasizing these rare yet successful trajectories via novelty-guided advantage shaping, the learner can expand its solution repertoire and avoid premature convergence to familiar but suboptimal reasoning patterns. However, exploiting trajectory-level novelty introduces several challenges: (i)

novelty must be defined in a way that distinguishes informative reasoning paths without rewarding spurious randomness; (ii) novelty-based signals must be calibrated relative to correctness to avoid destabilizing training; and (iii) the shaping mechanism must operate efficiently at scale without introducing additional rollout overhead.

To address these challenges, we next present how *XRPO* balances exploration and exploitation by accounting for runtime uncertainty across rollouts within and across rounds (§3.2), while simultaneously performing novelty-based advantage sharpening to exploit rich trajectory signals (§3.3).

### 3.2. *XRPO* Exploration on Valuable Prompts

GRPO's static rollout allocation ignores the inherent uncertainty of stochastic rollout sampling, leading to inefficient exploration. However, as each rollout is a stochastic draw from the model's latent token distribution, effective exploration requires reasoning explicitly about uncertainty and marginal information gain. In fact, an effective rollout allocation strategy should satisfy three desiderata: (i) *uncertainty awareness*, prioritizing rollouts that most reduce estimation error; (ii) *generalizability*, remaining agnostic to model architecture and reward scale; and (iii) *lightweight implementation*, avoiding costly inner-loop optimization during training.

**Hierarchical Rollout Planning.** We propose a novel hierarchical rollout strategy that models the priority of each prompt as a combination of both the expected reduction in uncertainty and an exploration bonus. Our design first prioritizes prompts with the best estimated reward uncertainty reduction. The uncertainty in the estimated mean reward $\mu_q$ can be quantified by the half-width of the Student's $t$-confidence interval:

$$h_q(n_q) = t_{1-\alpha/2,n_q-1} \frac{s_q}{\sqrt{n_q}}, \tag{1}$$

where $\bar{r}_q$ and $s_q$ are the sample mean and standard deviation of rewards for prompt $q$, $n_q$ is the number of rollouts, and $t_{1-\alpha/2,n_q-1}$ is the critical value of the $t$-distribution with $n_q - 1$ degrees of freedom at confidence $1 - \alpha$. Now the expected uncertainty reduction from one additional rollout can be approximated by

$$\hat{\Delta}_q(n_q) = h_q(n_q) - h_q(n_q + 1)$$
$$\approx s_q \left( \frac{t_{1-\alpha/2,n_q-1}}{\sqrt{n_q}} - \frac{t_{1-\alpha/2,n_q}}{\sqrt{n_q+1}} \right) \tag{2}$$

This term favors prompts where an additional rollout provides the most statistical reward benefit.

However, simply allocating more rollout budgets to the uncertain prompts will overlook the sparsely sampled and hard prompts. Hence, we shift some amount of rollout budgets toward sparsely sampled prompts by adding an exploration bonus, encouraging better exploration–exploitation trade-offs: $\phi_q(T, n_q) = \lambda \sqrt{\frac{\log(1+T)}{n_q}}$, where $T$ is the total number of rollouts allocated in the current round, and $\lambda > 0$ is a tunable hyperparameter that trades off uncertainty-driven exploitation against exploration.

As such, the final priority score for allocating the next rollout to prompt $q$ is

$$\Pi_q = \hat{\Delta}_q(n_q) + \phi_q(T, n_q). \tag{3}$$

To prevent cold-start issues and degenerate groupings, we adopt a phased rollout allocation strategy. Each prompt first receives $n_{\text{base}}$ rollouts to establish a baseline signal. For example, under a total budget of 128 rollouts, we partition the allocation into three rounds: the first 64 are uniformly distributed across prompts, while the remaining 64 are divided across the second and third phases, where each prompt receives rollouts in proportion to its current priority score $\Pi_q$. This phased design enables periodic re-estimation of both $\bar{r}_q$ and $s_q$, thereby stabilizing allocation dynamics over time. Consequently, our design preserves a balanced trade-off between reducing uncertainty for high-variance prompts and sustaining exploration on undersampled ones.

**Breaking Symmetry via ICL Seeding.** While our rollout planner prioritizes high-variance prompts, hard prompts in GRPO often remain stuck at zero reward (§3.1), forming a zero-reward symmetry that standard exploration and self-refinement (Ding et al., 2025) cannot break. We therefore introduce ICL into GRPO training: conditioning on verified successes from similar tasks injects informative structure into otherwise unsolved prompts, breaking this symmetry and enabling in-context policy improvement. We formulate our ICL seeding strategy as follows.

For any prompt $q$ whose rollouts all fail, we retrieve up to $K$ similar training questions with verified reward solutions from an evolving ICL corpus. The corpus is initialized as empty and is incrementally populated with successful rollouts collected during training. During each rollout allocation phase, prompts with no successful rollouts allocate their rollout budget to ICL seeding, while prompts with at least one success receive standard rollouts. We note that ICL seeding is not simply replay or data augmentation (Zhang et al., 2025a), which reinforces successful trajectories only for identical prompts and provides no benefit when a prompt has never received reward. ICL seeding transfers verified reasoning patterns across related problems, injecting structure into otherwise unsolved cases. In practice, the ICL corpus grows conservatively from baseline-correct solutions and high-quality rollouts without external teachers. It is worth noting that corpus freshness is maintained implicitly: only verified successful rollouts are stored, and the

pool is continuously updated with on-policy successes, so it naturally evolves with the policy without explicit pruning.

Our evaluations show that phased rollout allocation and ICL seeding introduce negligible overhead, as calculating the priority score, prefilling the exemplars are fast, and autoregressive rollout generation dominates per-step latency. Moreover, the seeding design accelerates training convergence, ultimately reducing end-to-end completion time (§4.2).

### 3.3. *XRPO* Exploitation of the Sampling Trajectories

While our rollout planner encourages exploration of valuable prompts, we observe that rule-based rewards (e.g., binary 0/1 signals) remain too sparse, often collapsing distinctions among diverse rollouts. This suppresses the rich information embedded in generation trajectories and drives the model toward homogenized, suboptimal behaviors. To overcome this limitation, we extend the classical entropy bonus (Williams, 1992; Mnih et al., 2016) from the token level to the sequence level, and instantiate it with a group-relative, novelty-aware advantage sharpening mechanism.

**Novelty-Guided Advantage Sharpening.** The concept of *novelty* could be intuitively interpreted as the extent to which a rollout deviates from the estimated entropy of the entire sequence trajectory space. Formally, in autoregressive models, token-level entropy is defined as $H_t = -\sum_{v \in \mathcal{V}} \pi_\theta(v \mid x, y_{<t}) \log \pi_\theta(v \mid x, y_{<t})$, while sequence-level entropy considers full trajectories: $H(\pi_\theta) = -\sum_{y \in \mathcal{Y}} \pi_\theta(y \mid x) \log \pi_\theta(y \mid x)$. Under autoregressive factorization, this becomes $H(\pi_\theta) = -\mathbb{E}_{y \sim \pi_\theta} \left[ \sum_{t=1}^{|y|} \log \pi_\theta(y_t \mid x, y_{<t}) \right]$. To estimate it in practice, we can define the length-normalized log-likelihood score for a sampled trajectory $y$ as

$$s(y) = \frac{1}{|y|} \sum_{t=1}^{|y|} \log \pi_\theta(y_t \mid x, y_{<t}), \tag{4}$$

and estimate the full trajectory space entropy with $|H(\pi_\theta)| \propto \frac{1}{N} \sum_{j=1}^{N} s(y_j) = \bar{s}$, which is the averaged length-normalized log-likelihood score $\bar{s}$ across $N$ sampled rollouts. Then the *novelty* of rollout $y_i$ is defined as

$$\eta_i = e^{s(y_i) - \bar{s}}, \tag{5}$$

where $\eta_i < 1$ indicates a more uncertain (i.e., novel) sequence relative to the group. This provides a direct, sequence-level measure of how atypical a rollout is under the model's own distribution.

We integrate this *novelty* into training by sharpening the advantage for rule-based rewards. Specifically, if rollout $y_i$ receives full reward (e.g., 1), we adjust its default GRPO advantage $A_i$ as,

$$A_i^+ = A_i + \min \left\{ \max \left\{ \lambda_{\text{nov}}(1 - \eta_i), 0 \right\}, \kappa_{\text{clip}} A_i \right\} \tag{6}$$

---

**Algorithm 1** *XRPO*: eXplore-eXploit GRPO

**Require:** LLM $\pi_\theta$, evaluator Eval, base rollouts $n_{\text{base}}$, allocation $n_r$, rounds $N_{\text{plan}}$, batch $\mathcal{Q}$, ICL corpus $\mathcal{C}_{\text{init}}$
**Ensure:** Rollouts $Y_{\text{comp}}$; Advantage $A^+$; updated corpus $\mathcal{C}$
1: $Y_{\text{comp}} \leftarrow \varnothing$; $A^+ \leftarrow \varnothing$; $\mathcal{C} \leftarrow \mathcal{C}_{\text{init}}$        ▷ init
2: **for** $q \in \mathcal{Q}$ **do**
3:     $Y_{\text{comp}}[q] \leftarrow \{y^{(i)} \sim \pi_\theta(\cdot \mid q)\}^{n_{\text{base}}}$   ▷ base rollouts
4: **end for**
5: **for** $t = 1, \ldots, N_{\text{plan}}$ **do**        ▷ planning rounds
6:     $S_{\mathcal{Q}} \leftarrow \{\text{Eval}(Y_{\text{comp}}[q])\}_{q \in \mathcal{Q}}$
7:     Alloc $\leftarrow$ STATALLOC$(S_{\mathcal{Q}}, n_r)$
8:     **for** $q \in \mathcal{Q}$ **do**
9:        $\text{acc}_q \leftarrow \mathbb{I}\{\exists y \in Y_{\text{comp}}[q] : \text{Eval}(y) = 1\}$
10:       $\tilde{q} \leftarrow$ ICLPROMPT$(q, \mathcal{C})$ **if** $\text{acc}_q{=}0$ **else** $q$
11:       $Y_t \leftarrow \{y \sim \pi_\theta(\cdot \mid \tilde{q})\}^{\text{Alloc}[q]}$   ▷ new rollouts
12:       $Y_{\text{comp}}[q] \leftarrow Y_{\text{comp}}[q] \cup Y_t$
13:     **end for**
14: **end for**
15: $A^+ \leftarrow$ ADVSHARP$(\text{Eval}, Y_{\text{comp}}, \pi_\theta)$     ▷ advantage
16: UPDATECORPUS$(\mathcal{C}, Y_{\text{comp}})$    ▷ add correct rollouts
17: **return** $Y_{\text{comp}}, A^+, \mathcal{C}$

---

where $\lambda_{\text{novelty}}$ controls how strong the novelty bonus is and $\kappa_{\text{clip}}$ caps the maximum bonus. Only rollouts with $\eta_i < 1$ (Refer Eq. 5) are boosted. We only shape rollouts receiving full reward because the binary ORM cannot reliably distinguish "almost-correct" failures, and boosting low-likelihood incorrect rollouts can amplify noisy signals and destabilize training, while reweighting within correct rollouts preserves the expected correctness reward. It's also worth noting that the shaping strategy is free of length bias (§ D.1). Our sharpening mechanism offers two benefits: (i) *Boundary Expansion*: By rewarding novel yet correct sequences, the policy boundary is pushed outward, improving generalization. (ii) *Dense Differentiation*: Groups with degenerated advantages gain additional reward signals, mitigating collapse and accelerating convergence.

Algorithm 1 summarizes how *XRPO* integrates exploration and exploitation in a cohesive loop. After initializing with a fixed number of base rollouts per prompt, *XRPO* proceeds in phased allocation rounds (Lines 5–14), where rollout allocations are distributed by jointly considering the expected reduction in statistical uncertainty and the need to continue sampling underexplored prompts. For prompts that consistently fail, *XRPO* activates ICL seeding by injecting contextual examples from the evolving corpus (Line 10), breaking zero-reward symmetry and enabling policy improvement. At the end of each phase, the algorithm collects new rollouts, updates prompt-level statistics, and repeats. Once all rollouts generation is complete we compute advantage and correct rollouts are further refined with novelty-guided advantage sharpening.

# 4. Experiment

## 4.1. Experimental Settings

Our method is implemented using the VERL pipeline (Sheng et al., 2024) and leverages vLLM (Kwon et al., 2023) for rollout execution. All experiments are conducted on 16× H200 GPUs (141GB each).

**Models & Datasets.** We evaluate our approach on Qwen3-1.7B (Yang et al., 2025a), Qwen2.5-7B-Instruct (Qwen et al., 2024), and Llama-3.2-3B (Grattafiori et al., 2024). Following existing advances (Zheng et al., 2025a), Qwen3-1.7B is configured with a maximum generation length of 16,384 tokens and an input context window of 8,192 tokens, with reasoning mode enabled. For Qwen2.5-7B-Instruct, we use a maximum generation length of 8,192 tokens and a prompt length up to 4,096 tokens. For Llama-3.2-3B, we use a maximum generation length of 2,048 tokens and allow prompt lengths up to 4,096 tokens.

Our evaluations cover two primary tasks across seven datasets: (1) *Math Reasoning*: AIME 2024/2025 (AIME), HMMT 2025 (Feb) (Balunović et al., 2025), BRUMO 2025 (Balunović et al., 2025), and MATH (Lightman et al., 2023); and (2) *Code Generation*: Codeforces (Quan et al., 2025) and LiveCodeBench v5 (LCBv5) (Jain et al., 2024).

**Training Setup.** We train using the AdamW optimizer with a learning rate of $2 \times 10^{-6}$, a batch size of 16, and a KL coefficient of $\beta = 0.001$. Our rollout strategy uses 4 base rollouts per prompt, with a total of 128 rollouts for Qwen3-1.7B and 64 for Qwen2.5-7B-Instruct, distributed across 2 dynamic rollout allocation rounds. For ICL seeding, we incorporate 2 solved exemplars generated by the same model and apply novelty-based advantage shaping with $\lambda_{\text{novelty}} = 2.5$ and $\kappa_{\text{clip}} = 0.5$. To construct the ICL corpus, problem similarity is computed using Qwen3-Embedding-8B (Zhang et al., 2025b). We further provide a sensitivity analysis to demonstrate *XRPO*'s consistent effectiveness (§4.3).

For all other experiments in the paper, we construct the training data by randomly sampling 10K examples from both the DAPO-Math-17k dataset (Yu et al., 2025a) and the DeepCoder-Preview-Dataset (Luo et al., 2025), keeping the dataset size practical. For the training convergence experiment, we use Qwen2.5-Math-1.5B (Yang et al., 2024) with a maximum generation length of 2048 tokens and a maximum prompt length of 1024 tokens. This model is trained entirely on the MATH dataset (Lightman et al., 2023).

**Baselines.** We compare *XRPO* against four advances:

- GRPO (Shao et al., 2024), which applies Token-Level Policy Gradient Loss and clipping with $\epsilon \in [0.2, 0.28]$;
- GSPO (Zheng et al., 2025a), which optimizes at the sequence level with $\epsilon \in [0.0003, 0.0004]$;

*Table 1.* **Qwen3-1.7B reasoning performance.** *XRPO* outperforms GSPO, DAPO (DS), and TreePO Sampling (TPO-S) across math and code benchmarks.

*(a)* Math reasoning performance on AIME'25, HMMT'25, and BRUMO'25

| Method | Metric | AIME'25 | HMMT'25 | BRUMO'25 | Avg. |
|--------|--------|---------|---------|----------|------|
| GSPO | pass@1 | 33.23 | 21.25 | 45.20 | 33.23 |
|  | cons@32 | 46.67 | 20.00 | 56.67 | 41.11 |
| DAPO | pass@1 | 32.50 | 20.42 | 39.90 | 30.94 |
|  | cons@32 | 33.33 | 23.33 | 43.33 | 33.33 |
| TPO-S | pass@1 | 26.04 | 17.29 | 37.40 | 26.91 |
|  | cons@32 | 33.33 | 20.00 | 46.67 | 33.33 |
| *XRPO* | pass@1 | **35.72** | **22.29** | **47.39** | **35.13** |
|  | cons@32 | **50.00** | **26.67** | **60.00** | **45.56** |

*(b)* Performance on MATH and Codeforces.

| Method | MATH | Codeforces | Avg. |
|--------|------|------------|------|
| GSPO | 90.33 | 13.72 | 52.03 |
| DAPO | 89.33 | 9.48 | 49.41 |
| TPO-S | 84.65 | 9.51 | 47.08 |
| *XRPO* | **90.54** | **13.80** | **52.17** |

*Table 2.* **Llama-3.2-3B performance.** *XRPO* outperforms GRPO across math benchmarks.

| Method | Metric | AIME'24 | AIME'25 | BRUMO'25 | Avg. |
|--------|--------|---------|---------|----------|------|
| GRPO | pass@1 | 7.19 | 0.52 | **4.79** | 4.17 |
|  | pass@4 | 16.17 | 1.97 | 7.82 | 8.65 |
| *XRPO* | pass@1 | **9.27** | **0.63** | 3.54 | **4.48** |
|  | pass@4 | **18.70** | **2.38** | **9.48** | **10.19** |

- DAPO (Yu et al., 2025a) with three of its four components: Clip-Higher ($\epsilon \in [0.2, 0.28]$), Dynamic Sampling, and Token-Level Policy Gradient Loss;
- TreePO sampling strategy from TreePO (Li et al., 2025c), following the settings in their paper.

**Evaluation Metrics.** Following (Wang et al., 2025b), we evaluate models every 30 training steps on the validation set. For inference, we generate $n = 32$ candidate solutions per problem using nucleus sampling (Temperature = 1.0, top-$p = 0.95$ for Qwen models and Temperature = 0.6, top-$p = 0.9$ for Llama-3.2-3B) and measure performance with `pass@k` (probability that at least one of the top-$k$ samples is correct) and `cons@32` (fraction of problems with correct majority-vote accuracy aggregated using 32 sampled solutions per problem). It's worth noting that, during evaluation we do not use the ICL corpus at all, which means all methods, including *XRPO*, are evaluated with standard prompting only, without retrieval or in-context augmentation, eliminating any potential data leakage concerns.

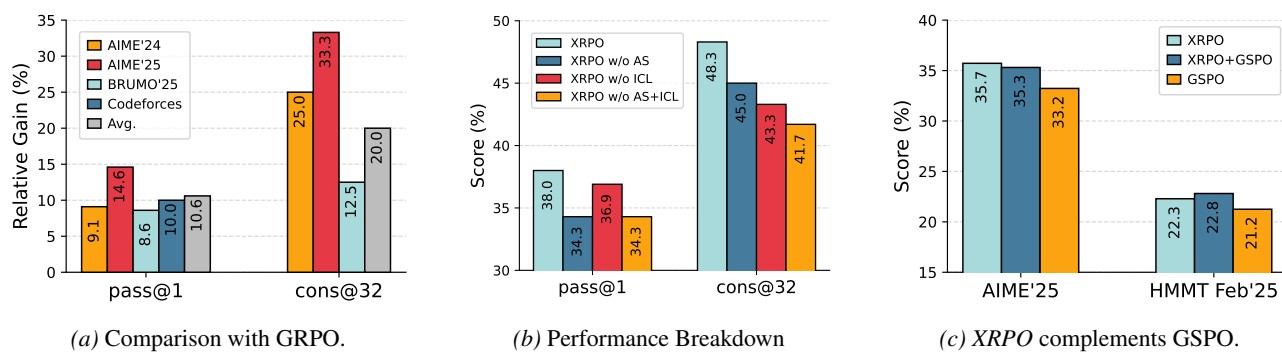

*(a)* Comparison with GRPO.    *(b)* Performance Breakdown    *(c)* *XRPO* complements GSPO.

*Figure 3.* *XRPO* improves performance across multiple dimensions.

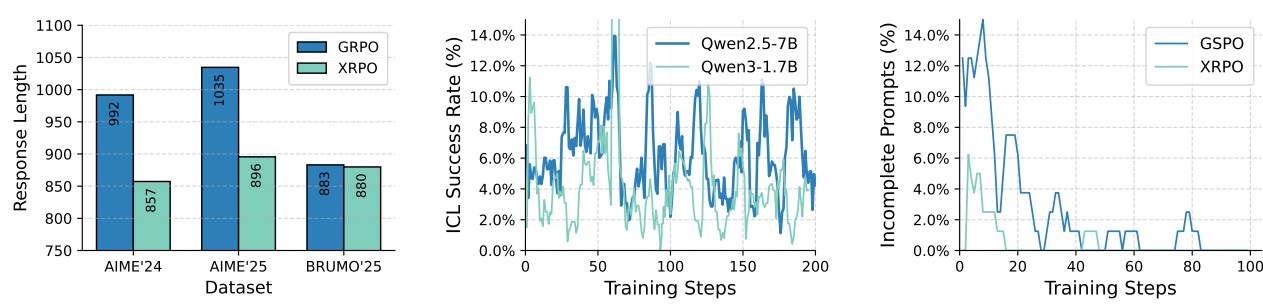

*(a)* *XRPO* achieves better inference efficiency.

*(b)* *XRPO* maintains a consistently stable ICL success rate.

*(c)* *XRPO* achieves better response completion rates.

*Figure 4.* Break down analysis on *XRPO*.

## 4.2. End-to-end Performance

We begin with an end-to-end analysis of efficiency and model quality to demonstrate the effectiveness of our prompt exploration and reward exploitation strategies.

***XRPO* improves model quality.** Table 1, Figure 3a and Table 2 show that *XRPO* consistently outperforms state-of-the-art baselines across challenging math reasoning and code generation benchmarks. For Qwen2.5-7B-Instruct, *XRPO* delivers substantial relative improvements of +9.2% in pass@4 and +20% in cons@32. For the complete results and comparisons with all baselines, please refer to Appendix A.2. On Llama-3.2-3B, *XRPO* still achieves a higher average performance despite the significant difficulty of AIME25. For Qwen3-1.7B (with reasoning mode enabled), *XRPO* achieves a +2.5% improvement in average cons@32 and +1.4% in pass@1 over GSPO across datasets, demonstrating gains in both raw accuracy and output consistency. This improvement is particularly notable given that GSPO is among the most recent and competitive approaches (Zheng et al., 2025a). Further, Qwen3 is already a reasoning-focused model and challenging to train due to its heavily trained nature; thus, observing improvements highlights the practical value of *XRPO*. Moreover, *XRPO* not only enhances accuracy but also delivers superior train-

ing and inference efficiency (see later). We further evaluate *XRPO* at multiple decoding budgets, showing consistent improvements across all $k$ values (Appendix H).

***XRPO* achieves faster training convergence.** Beyond accuracy improvements, our experiment shows that *XRPO* achieves noticeable training efficiency speedup on Qwen2.5-Math-1.7B when trained on the MATH training dataset. Specifically, it reaches 82.5% accuracy on GSM8K by step 420, whereas GRPO requires approximately 1K steps to achieve similar performance, yielding a speedup of $2.4\times$. Similarly, on MATH-500 (Lightman et al., 2023), *XRPO* attains 75% accuracy by step 450, compared to GRPO's about 1.2K steps, corresponding to a $2.7\times$ speedup. These findings validate the effectiveness of our exploration–exploitation design: by prioritizing edge-policy cases, the model quickly absorbs salient information and effectively expands its decision boundaries. Detailed training dynamics curves are provided in Appendix I.

***XRPO* introduces negligible training overhead.** *XRPO* only introduces limited overhead and could operate at comparable latency compared to the vanilla GRPO algorithm. We evaluated the per-step end-to-end latency using a batch size of 64, 256 rollouts per prompt, and two dynamic rounds to better simulate common training settings. Our results

*Table 3.* ICL reduces average generation length for both Qwen3 and Qwen2.5.

| Response Length | Qwen2.5 | Qwen3 |
|---|---|---|
| Without ICL | 881.7 | 12906.1 |
| With ICL | 827.2 | 8427.9 |
| Reduction | 6.17% | 34.7% |

*Table 4.* Hyperparameter sensitivity analysis.

| *(a)* Novelty sharpening. | | | | *(b)* Rollout planning. | | |
|---|---|---|---|---|---|---|
| $\lambda_{novelty}$ | $\kappa_{clip}$ | Score | | $\lambda$ | $\alpha$ | Score |
| 2.5 | 0.5 | 53.66 | | 0.10 | 0.95 | 53.66 |
| 5.0 | 0.5 | 54.94 | | 0.12 | 0.93 | 53.06 |
| 1.0 | 0.5 | 56.26 | | 0.08 | 0.95 | 56.27 |
| 2.5 | 0.8 | 55.24 | | 0.12 | 0.97 | 55.32 |
| | | | | 0.08 | 0.93 | 55.89 |
| avg. | | 55.0±1.1 | | avg. | | 54.8±1.4 |

indicate that the per-step latency ratio between *XRPO* and baseline GRPO is about 1.047, mere 4.7% overhead. The ICL corpus is loaded once at the start of training (4.22 seconds), advantage-shaping introduces nearly no overhead (0.329 seconds), and constructing ICL prompts is extremely fast ($< 0.001$ seconds per prompt) due to simple hash-map lookups over precomputed neighbors and responses.

We further provide a theoretical latency analysis in Appendix B. This analysis shows that the overhead introduced by dynamic rollout allocation becomes negligible when the rollout workload is large, while remaining bounded in the worst case. Overall, *XRPO* not only reduces the number of convergence steps but keeps near-identical per-step latency.

**XRPO achieves better inference efficiency.** Table 3 reports that *XRPO* consistently generates shorter and more efficient solution paths, emerging naturally from improved exploration under context constraints. *XRPO* guides exploration toward solution paths that are likely to remain valid within the context window via the Rollout Allocator and In-Context Seeding. Since successful solutions must fit within that window, *XRPO* implicitly favors trajectories that are concise enough to be complete. This interpretation is supported by empirical results in Table 3, which show that in-context learning reduces average generation length by 34.7 percent for Qwen3 and 6.17 percent for Qwen2.5. Figure 4a further shows that *XRPO* reduces average response length by 13.6% on AIME'24 and 13.4% on AIME'25 relative to GRPO, indicating more targeted reasoning with fewer steps. Moreover, we observe no example truncation in our training experiments after applying ICL seeding, as the inclusion of exemplars often leads the model to produce shorter responses. Moreover, even when exemplars are truncated, our ablation studies show that ICL seeding continues to improve model performance (see Appendix D.2 for details).

### 4.3. Ablation Study and Sensitivity Analysis

**Performance Breakdown by Design Components.** We ablate *XRPO* into four variants to isolate the contribution of each component: (1) XRPO *w/o ICL*, which removes ICL Seeding (no symmetry-breaking); (2) XRPO *w/o AS*, which disables Advantage Sharpening (no trajectory-level adjustment); (3) XRPO *w/o (AS+ICL)*, which removes both ICL Seeding and Advantage Sharpening, leaving rollout

planning as the sole training signal; and (4) XRPO *w/o HRP*, which replaces hierarchical rollout planning with uniform allocation while retaining ICL and AS.

Figure 3b summarizes results on Qwen3-1.7B (reasoning mode) averaged across AIME'24, AIME'25, HMMT'25, and BRUMO'25. Removing any module consistently degrades performance, highlighting their integrity to *XRPO*. Dropping ICL causes a 4.2% decline in cons@32 performance, highlighting the importance of symmetry-breaking. Disabling AS reduces sample efficiency and weakens generalization, leading to 3.5% accuracy degradation in pass@1. While rollout planning alone provides a strong baseline, only the full combination of RP, ICL, and AS delivers the best performance. Removing HRP (w/o HRP) drops pass@1 from 37.81 to 35.10 and cons@32 from 47.50 to 45.83, confirming that HRP contributes both standalone and integrated value, independently outperforming DAPO. For dataset-wise comparison, see Appendix A.1.

We further analyze how ICL contributes to better model convergence. Figure 4b shows that with ICL, *XRPO* correctly flips hard questions by an average of 6.2% and 4.2% on Qwen2.5-7B-Instruct and Qwen3-1.7B, respectively, demonstrating its effectiveness in breaking the edge of model capability. Similarly, Figure 4c indicates that ICL largely reduces the fraction of prompts that fail to produce complete responses within the 16K context length across all rollouts. These directly illustrate ICL's role in enabling the model to tackle problems beyond its decision boundary. We further conduct ablations on key ICL design choices, including train/test-time usage and retrieval quality, and observe that the gains mainly stem from training-time adaptation with strong semantic retrieval, while even random ICL remains beneficial. Detailed results are deferred to Appendix E and Appendix F.

**Robustness to Hyperparameters.** To evaluate the robustness of our method to hyperparameter choices, we conduct experiments on Qwen2.5-1.5B with results on the MATH-500 dataset (Lightman et al., 2023) by varying both the novelty bonus $\lambda_{novelty}$ and the clamp factor $\kappa_{clip}$, as well as the hyperparameters related to hierarchical rollout planning,

including the exploration strength $\lambda$ and the confidence interval $\alpha$. Results are shown respectively in Table 4a, and Table 4b. Overall, performance remains stable across a broad range of settings. When varying $\lambda_{\text{novelty}}$ and the clamp factor $\kappa_{\text{clip}}$, accuracies fluctuate only slightly, with a mean of 55.02% and a sample standard deviation of 1.07. As for $\lambda$ and $\alpha$, the accuracies fluctuate within a narrow band with a mean of 54.84% and a sample standard deviation of 1.41. These findings confirm that our design is resilient to hyperparameter variations, ensuring reliable performance without requiring extensive tuning.

**Compatibility with Other SOTA Methods.** Our design is complementary to recent optimization advances and can be seamlessly integrated with state-of-the-art methods. As shown in Figure 3c, *XRPO* achieves 2.5% better accuracy than GSPO when applied on Qwen3-1.7B. Importantly, *XRPO* remains fully compatible and yields comparable or improved performance when paired with GSPO. These results highlight that our exploration–exploitation mechanisms complement GSPO's optimization strategy, further enhancing downstream reasoning quality.

## 5. Conclusion

This paper introduces *XRPO*, a principled RLHF framework that rebalances exploration and exploitation in rollout optimization. By introducing hierarchical rollout planning that prioritizes high-variance prompts near decision boundaries, ICL seeding that breaks zero-reward symmetry on hard problems, and novelty-aware advantage sharpening that amplifies low-probability yet correct responses, XRPO pushes models beyond their current capability limits. Extensive experiments demonstrate consistent improvements over GRPO and recent advances, with over 1.4% higher accuracy, 2.5% higher average consistency across math and coding benchmarks, and 2.7$\times$ faster convergence.

**Limitations and Future Directions.** Due to resource constraints, we have not evaluated *XRPO* on very large-scale models (e.g., 300B+ parameters), and broader validation on diverse tasks beyond math reasoning and code generation is left for future work. Additionally, our ICL corpus is currently populated from the model's own successful rollouts; augmenting it with stronger teacher-generated responses could provide richer reasoning scaffolds for harder problems and further extend the exploration frontier.

## Acknowledgements

We thank the anonymous reviewers for their constructive and insightful feedback. This work was supported in part by gifts from Amazon, Cisco, and Google, and by an award from NVIDIA Academic Program. It also utilized the Delta system at the National Center for Supercomputing Applications (NCSA) through allocation CIS240236 from the ACCESS program.

## Impact Statement

This work aims to advance the field of machine learning by improving modeling and training methodologies. We do not foresee significant negative societal impacts arising from this work beyond those commonly associated with general-purpose machine learning research.

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

# Appendix

# A. Additional Experimental Results

### A.1. Ablation Study and Baseline Comparison

The results in Table 5 show that the ablated *XRPO* variants consistently outperform the other sampling baselines. Even when AS, ICL, or both are removed, these reduced versions still achieve higher pass@1 and cons@32 scores than DAPO and TreePO Sampling across most datasets. This indicates that the underlying *XRPO* framework remains strong even without its individual components.

*Table 5.* **XRPO ablations vs. sampling baselines.** Removing advantage sharpening (AS) or ICL seeding degrades performance, yet ablated variants still outperform DAPO and TreePO Sampling.

| Method | Metric | AIME'24 | AIME'25 | HMMT'25 | BRUMO'25 | Avg. |
|---|---|---|---|---|---|---|
| XRPO | pass@1 | **46.46** | 35.72 | 21.66 | **47.39** | **37.81** |
| | cons@32 | **56.66** | **50.00** | 23.33 | **60.00** | **47.50** |
| XRPO w/o AS | pass@1 | 39.17 | 33.13 | 20.10 | 45.00 | 34.35 |
| | cons@32 | 53.33 | 40.00 | **26.67** | 60.00 | 45.00 |
| XRPO w/o ICL | pass@1 | 44.58 | **35.94** | 20.94 | 46.04 | 36.87 |
| | cons@32 | 53.33 | 43.33 | 20.00 | 56.67 | 43.33 |
| XRPO w/o AS+ICL | pass@1 | 40.72 | 31.45 | 19.06 | 46.04 | 34.32 |
| | cons@32 | 46.66 | 40.00 | 20.00 | 60.00 | 41.67 |
| DAPO | pass@1 | 40.31 | 32.50 | 20.42 | 39.90 | 33.28 |
| | cons@32 | 43.33 | 33.33 | 23.33 | 43.33 | 35.83 |
| TreePO Sampling | pass@1 | 38.33 | 26.04 | 17.29 | 37.40 | 29.77 |
| | cons@32 | 50.00 | 33.33 | 20.00 | 46.67 | 37.50 |

### A.2. Results on Qwen2.5-7B-Instruct

In Table 6 we present expanded evaluation results for Qwen2.5-7B-Instruct across the AIME and BRUMO benchmarks.

# B. Latency Anlysis on *XRPO*

Let $N$ denote the total rollout budget, $m_p$ the system parallelism, and $t_0$ the per-rollout execution time. The baseline uniform allocation requires

$$T_1 = \left\lceil \frac{N}{m_p} \right\rceil t_0 = \frac{N + m_p - 1}{m_p} t_0. \tag{7}$$

Assuming we set the dynamic rounds to $n$, which divides $N$. Including the planning overhead per-round, $\Delta t$, the total runtime will be

$$T_2 = n \left( \left\lceil \frac{N/n}{m_p} \right\rceil t_0 + \Delta t \right)$$
$$= \frac{N + nm_p - n}{m_p} t_0 + n\Delta t. \tag{8}$$

Since $\Delta t$ involves only lightweight statistics (e.g., computing reward variances), it is negligible compared to rollout

*Table 6.* **Qwen2.5-7B-Instruct performance.** *XRPO* outperforms GRPO and sampling baselines across math benchmarks.

| Method | Metric | AIME'24 | AIME'25 | BRUMO'25 | Avg. |
|---|---|---|---|---|---|
| GRPO | pass@1 | 10.31 | 6.73 | 15.83 | 10.96 |
| | pass@4 | 18.10 | 15.31 | 26.30 | 19.90 |
| | cons@32 | 13.33 | 10.00 | 26.67 | 16.67 |
| DAPO | pass@1 | 10.52 | 6.67 | 17.92 | 11.70 |
| | pass@4 | 20.22 | 15.57 | 28.72 | 21.50 |
| | cons@32 | 16.67 | 10.00 | 26.67 | 17.78 |
| TreePO Sampling | pass@1 | 8.96 | 5.10 | **19.17** | 11.08 |
| | pass@4 | 17.20 | 13.72 | **29.55** | 20.16 |
| | cons@32 | 16.67 | 6.67 | 26.67 | 16.67 |
| XRPO | pass@1 | **11.25** | **7.71** | 17.19 | **12.05** |
| | pass@4 | **21.17** | **16.80** | 29.25 | **22.41** |
| | cons@32 | **16.67** | **13.33** | **30.00** | **20.00** |
| *Rel. Gain wrt GRPO* | pass@1 | +9.1% | +14.6% | +8.6% | +10.7% |
| | pass@4 | +17.0% | +9.7% | +11.2% | +12.6% |
| | cons@32 | +25.0% | +33.3% | +12.5% | +23.6% |

execution. The latency ratio can be rewritten explicitly as

$$\frac{T_2}{T_1} = \frac{N + nm_p - n}{N + m_p - 1}$$
$$= n - \frac{(n-1)N}{N + m_p - 1} \tag{9}$$
$$= n - \frac{(n-1)}{1 + \frac{m_p - 1}{N}}$$

If the total batch size $N$ is far greater than the degree of parallelism $m_p$ (*i.e.* $\lim \frac{m_p}{N} \to 0$), the dynamic allocator introduces no overhead. It is shown as

$$\frac{T_2}{T_1} = n - (n-1) = 1 \tag{10}$$

If we consider the worst case when $N$ is significantly small compared to $m_p$ (*i.e.* $\lim \frac{m_p}{N} \to \infty$), which is very unlikely to happen, we show that $T_2$ is still bounded as

$$\frac{T_2}{T_1} = n - 0 = n \tag{11}$$

Therefore, we could safely conclude that the overhead for dynamic allocation is negligible for large $N$, which is the typical case for both training and deployment, while still remaining bounded in the worst case.

# C. Breaking Symmetry via ICL Seeding

To address the challenge of systematically unsolved prompts, we integrate a few-shot in-context learning (ICL) seeding strategy into *XRPO* training. Whenever a given prompt fails to produce any successful solution in its base rollouts, the remaining rollout budget is allocated to an ICL-augmented prompt. These ICL prompts incorporate verified solved examples drawn from an evolving corpus of problem–solution pairs that the model has successfully answered in prior training steps. The similarity search for retrieval is conducted

using Qwen3-Embedding-8B (Zhang et al., 2025b), ensuring that only the most semantically relevant solved problems are selected as demonstrations. We limit the number of retrieved examples to $K = 2$ in order to conserve context length while still providing sufficient guidance. If no suitable solved examples exist, the prompt falls back to its zero-shot form.

The prompt template is structured into three components: (i) a `<task>` section containing general instructions, (ii) an `<examples>` block containing up to two similar problems and their corresponding verified solutions, and (iii) the new problem to be solved, formatted within a `<new_problem>` tag. The model is instructed to extract a general strategy from the examples, reason through the new problem, and finally output the answer in a standardized format (e.g., \boxed{} for mathematics or fenced code blocks for programming). To ensure feasibility within the model's context window, overly long example solutions are truncated as needed.

**Prompt Template.** ICL prompt used in our experiments is shown below:

```
<task>
 You are given several worked examples,
 each with a <problem> and a <solution>.
 Extract a general strategy, then think
 through the new problem, and finally
 provide the detailed solution.
</task>

<examples>
 <example id="1">
  <problem>[Example problem 1]</problem>
  <solution>[Correct solution 1]</solution>
 </example>
 <example id="2">
  <problem>[Example problem 2]</problem>
  <solution>[Correct solution 2]</solution>
 </example>
</examples>

<new_problem>[New problem]</new_problem>
```

## D. Context Length Analysis

### D.1. Novelty-Guided Advantage Sharpening is Free of Length Bias

We analyze whether the novelty-guided advantage sharpening mechanism introduces any preference for longer or shorter responses. Two observations confirm that the mechanism is free of such bias.

First, Equation 4 shows that each trajectory's log-likelihood score is normalized by its own length. This removes systematic preference for either long or short trajectories.

*Table 7.* **Length statistics of shaped entries.** Shaped entries show no length bias relative to correct rollouts or full groups.

| Metric | Correct Rollouts | Full Groups |
|---|---|---|
| Z-score | 0.257 | -0.280 |
| Relative Ratio | 1.040 | 0.991 |

*Table 8.* Performance difference between truncated vs. untruncated successful examples.

| Truncation (both samples) | Accuracy |
|---|---|
| 0% | 41.5 |
| 5% | 40.8 |
| 10% | 39.8 |
| 20% | 39.0 |
| No ICL | 37.7 |

Second, we examine the length distribution of shaped entries by computing their response-length z-scores and their relative ratios within full groups and within the sets of correct rollouts in those groups (Table 7). The results indicate that shaped entries lie within 0.25 standard deviations of the mean and are more than 99 percent close to the group-average length. These findings show no observable length bias introduced by the novelty score or the shaping mechanism.

### D.2. Impact of Truncation on ICL Seeding

We agree that truncation of long, multi-step exemplars can, in principle, remove critical reasoning steps and therefore degrade the quality of the ICL seed used during rollouts. But it's rarely observed during our training.

**Fraction of Truncated ICL Examples.** During the construction of the ICL corpus, we include only fully correct sampled solutions. These correct solutions are substantially shorter than failed or partially correct chains of thought. The mean length of a correct example in our corpus is approximately 900 tokens (Table 9). Because each ICL prompt includes only two retrieved exemplars, the combined length of the exemplars plus the query remains comfortably within the allowable context window. As a result, we did not observe any truncation of ICL exemplars during training.

**Effect of Truncation on Performance.** To directly assess the impact of truncation, we performed an ablation in which we artificially truncated long ICL exemplars and compared rollout performance against their full-length versions. As shown in Table 8, truncation leads to a performance drop of 0.7% and 1.7% when both ICL samples are truncated by 5% and 10%, respectively, consistent with our intuition. However, such cases are extremely rare in our actual train-

*Table 9.* Training statistics.

| Model | ICL corpus | | Mean prompt length | Max prompt (w/ ICL) | Max allowed length |
|---|---|---|---|---|---|
| | Mean | P95 | | | |
| Qwen3 | 879.9 | 1,395 | 998 | 5,233 | 8,192 |
| Qwen2.5 | 711.2 | 1,219 | – | 3,826 | 4,096 |

*Table 10.* **ICL train/test ablation on Qwen2.5-7B-Instruct (pass@1).** Training-time ICL is sufficient; test-time ICL is redundant.

| Method | AIME'24 | AIME'25 | BRUMO'25 | Avg. |
|---|---|---|---|---|
| GRPO | 10.31 | 6.73 | 15.83 | 10.96 |
| *XRPO* | **11.25** | **7.71** | **17.19** | **12.05** |
| GRPO + ICL@test | 10.78 | 6.04 | 16.38 | 11.07 |
| *XRPO* + ICL@test | 11.31 | 6.98 | 17.01 | 11.77 |

*Table 11.* **Retrieval quality ablation (accuracy on DAPO-Math-17k).** Stronger retrieval yields larger ICL gains; even random ICL outperforms no ICL.

| Retrieval Strategy | Qwen2.5-7B | Qwen3-1.7B |
|---|---|---|
| Qwen3-Embed-8B (strong) | **24.5** | **41.5** |
| Qwen3-Embed-0.6B (weak) | 21.23 | 39.35 |
| Random | 20.32 | 38.12 |
| No ICL | 18.6 | 37.7 |

*Table 12.* ***XRPO* vs. CurES on Qwen3-1.7B.** Uncertainty-based allocation outperforms difficulty-based allocation.

| Method | Metric | AIME'25 | HMMT'25 | BRUMO'25 | Avg. |
|---|---|---|---|---|---|
| CurES | pass@1 | 33.13 | 21.15 | 41.67 | 31.98 |
| | cons@32 | 36.67 | 20.00 | 40.00 | 32.22 |
| *XRPO* | pass@1 | **35.72** | **22.29** | **47.39** | **35.13** |
| | cons@32 | **50.00** | **26.67** | **60.00** | **45.56** |

ing setup, where we observe 0% truncation, as shown in Table 9. Even under a truncation rate of 20%, we see that ICL consistently outperforms the baseline.

## E. ICL Train-Time vs. Test-Time Ablation

We evaluate four ICL configurations on Qwen2.5-7B-Instruct to determine whether ICL is needed at test time. For test-time ICL, we retrieve the top-2 most similar solved training examples via Qwen3-Embedding-8B (same as training) and prepend them as in-context demonstrations. As shown in Table 10, *XRPO* without test-time ICL achieves the best average (12.05), outperforming *XRPO* +ICL@test (11.77), GRPO+ICL@test (11.07), and GRPO (10.96). This confirms that the model internalizes reasoning strategies during training and test-time ICL is redundant. The slight drop with test-time ICL is expected: the ICL corpus contains training-level problems that are simpler than competition benchmarks, introducing a distributional mismatch that consumes context without providing useful guidance.

## F. ICL Retrieval Quality Ablation

We ablate retrieval quality using three strategies: Qwen3-Embedding-8B (strong, default), Qwen3-Embedding-0.6B (weak, same family), and random retrieval. We evaluate on DAPO-Math-17k prompts with 12 generations per prompt on two models. As shown in Table 11, strong retrieval outperforms all alternatives on both models (+5.9% over no ICL on Qwen2.5-7B, +3.8% on Qwen3-1.7B). Even random ICL provides benefit over no ICL (+1.7% and +0.4%), confirming that exposure to any correct reasoning pattern aids generation, while high-quality semantic matching is critical for full effectiveness.

## G. Comparison with CurES

We compare *XRPO* against CurES, a difficulty-based dynamic allocation method, on Qwen3-1.7B. As shown in Table 12, *XRPO* outperforms CurES on all benchmarks (avg. pass@1: 35.13 vs. 31.98; avg. cons@32: 45.56 vs. 32.22), empirically validating the uncertainty-based allocation over difficulty-based alternatives. CurES requires precomputing prompt difficulty and Bayesian posterior updates before training, whereas *XRPO* performs on-the-fly allocation using current reward uncertainty.

## H. pass@$k$ at Multiple Decoding Budgets

Table 13 reports pass@$k$ for Qwen3-1.7B at $k \in \{4, 8, 16, 32\}$ averaged across AIME'25, HMMT'25, and BRUMO'25. *XRPO* outperforms all baselines at every $k$, with the largest advantage at small-to-mid $k$ (+6.28 over DAPO at pass@8, +4.47 at pass@16) and remaining positive at pass@32, demonstrating improved inference scaling rather than only shifting performance at one decoding budget.

## I. Training Dynamics

*XRPO* achieves substantially faster convergence than GRPO. Figure 5a shows that on GSM8K, *XRPO* reaches 82.5% ac-

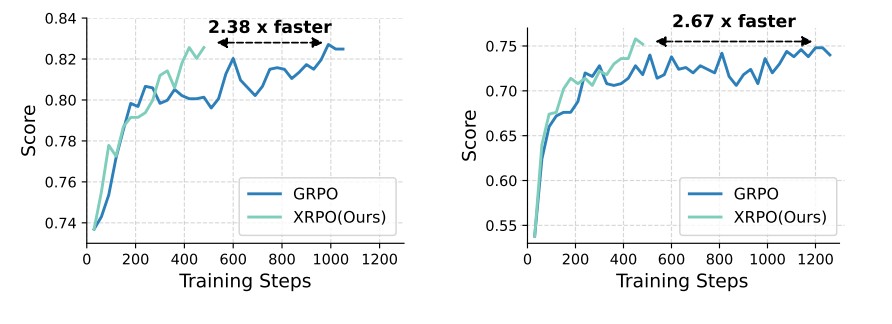
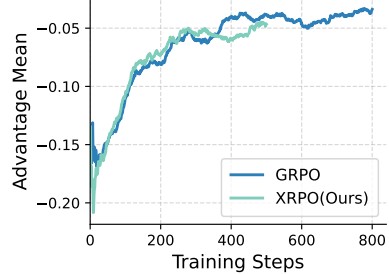

*(a)* Training accuracy on GSM8K.    *(b)* Training accuracy on MATH-500.    *(c)* Advantage convergence.

*Figure 5.* **Training dynamics.** *XRPO* reaches target accuracy 2.4–2.7× faster than GRPO with negligible per-step overhead.

*Table 13.* **pass@$k$ results on Qwen3-1.7B (averaged over AIME'25, HMMT'25, BRUMO'25).** *XRPO* outperforms all baselines across decoding budgets.

| Method | pass@4 | pass@8 | pass@16 | pass@32 |
|--------|--------|--------|---------|---------|
| GSPO | 46.09 | 52.01 | 57.60 | 62.22 |
| DAPO | 42.37 | 47.50 | 52.14 | 56.67 |
| TPO-S | 40.18 | 47.16 | 54.69 | 63.33 |
| *XRPO* | **48.65** | **55.92** | **62.07** | **65.56** |

curacy at step 420, whereas GRPO requires approximately 1K steps (2.4× speedup). Figure 5b shows similar trends on MATH-500: *XRPO* attains 75% accuracy at step 450 vs. GRPO's ∼1.2K steps (2.7× speedup). Figure 5c further shows that *XRPO*'s advantage metric converges earlier, while per-step overhead remains at only ∼4.7%.

## J. Non-Uniform Rollout Allocation Statistics

To directly measure exploration behavior, we analyze the rollout allocation distribution produced by HRP during training. HRP produces non-uniform allocations in 91.6% of training rounds, with per-prompt budgets ranging from 0 to 7 extra rollouts. High-uncertainty prompts consistently receive up to 7× more rollouts than the uniform baseline (Figure 6), confirming active exploration redirection throughout training. This validates that HRP meaningfully concentrates compute on informative prompts rather than distributing it uniformly.

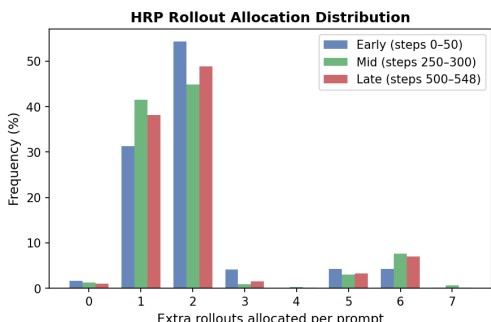

*Figure 6.* Distribution of per-prompt rollout allocations across training rounds. HRP produces non-uniform allocations in 91.6% of rounds.

