# OpenReview forum: "XRPO: Pushing the Limits of GRPO with Targeted Exploration and Exploitation"
_ICML.cc/2026/Conference — ICML 2026 regular_

### Official Review · Reviewer_yCVK · 2026-03-09

**Soundness:** 3
**Presentation:** 3
**Significance:** 3
**Originality:** 3
**Overall Recommendation:** 5
**Confidence:** 4

**Summary:**

- RL-LLM approaches suffer from limited exploration, especially on challenging prompts, and often rely on static rollout allocation across prompts together with sparse rewards
- XRPO conducts adaptive rollout allocation to prioritize prompts based on estimated uncertainty reduction
- XRPO uses in-context learning (ICL) seeding with retrieved solved exemplars to improve rollouts on challenging problems
- XRPO amplifies low-probability correct responses

**Compliance With Llm Reviewing Policy:**

Affirmed.

**Final Justification:**

This paper studies an important problem in RL for LLMs, namely how to better balance exploration and exploitation during rollout-based training. The proposed XRPO framework combines three components: hierarchical rollout allocation, ICL-based seeding for zero-reward prompts, and novelty-guided advantage shaping. Overall, I find the problem well-motivated, the methodology clearly presented, and the empirical results reasonably strong, showing consistent improvements over competitive baselines.

In terms of originality, while each individual idea is related to existing directions (e.g., dynamic sampling, in-context learning, and reward shaping), the particular combination and instantiation in XRPO are novel and well-integrated. The work makes a meaningful contribution to improving training efficiency and performance in RLVR settings.

Regarding soundness, my initial concerns focused on (1) the lack of direct exploration measurements, (2) the absence of an ablation isolating hierarchical rollout allocation, and (3) the unclear justification for the length normalization used in Equation (4). The rebuttal addresses several of these points. In particular, the authors provide a clear ablation for hierarchical rollout planning, which strengthens the empirical evidence and resolves a key concern. They also add additional analyses related to exploration (e.g., flip rates and rollout allocation behavior), which provide useful supporting evidence, although these remain indirect proxies rather than standard exploration metrics. The explanation for length normalization is helpful and improves intuition, though it is still somewhat heuristic and could benefit from a more formal derivation.

On the empirical side, the additional pass@k results and clarifications regarding dataset aggregation improve confidence in the reported gains. The results suggest that XRPO consistently improves both accuracy and inference scaling behavior across multiple settings.

Overall, the rebuttal meaningfully improves my confidence in the work by addressing the most critical gaps, particularly the missing ablation and empirical clarifications. While some concerns remain around theoretical grounding and direct measurement of exploration, these are relatively minor and do not undermine the core contributions. Balancing the strengths in significance, empirical performance, and clarity against the remaining weaknesses, I recommend acceptance.

**Key Questions For Authors:**

- Is there a justification for using the half-width of the Student’s $t$-confidence interval as the uncertainty metric?
- Is there any derivation or justification for the length-normalization in equation (4) for why it estimates $H(\pi_\theta)$ correctly?
- Are there any pass@k results for performance in Table 1, especially for different k values? It would be interesting to see the inference scaling performance of XRPO, especially since the paper claims that it induces more exploration.
- Are there cons@32 results for Table 1(b) as well, similar to Table 1(a)?
- Is there an ablation on what happens when you remove Hierarchical Rollout Exploration?
- Are there any results directly measuring the exploration induced?
- Figure 3b appears to average results over a different set of datasets than those reported in Table 1a. Could the authors clarify the exact datasets and aggregation used for this figure?

**Limitations:**

Yes

**Strengths And Weaknesses:**

## Soundness
- It is unclear where the length normalization comes from in equation (4)
- The paper claims improved exploration through adaptive rollout allocation and ICL seeding, but the empirical section does not include direct measurements of exploration (e.g., rollout diversity, entropy, or trajectory coverage). Instead, the evidence relies on downstream task accuracy and proxy statistics such as prompt flip rates. It is therefore difficult to determine whether the improvements arise from better exploration, improved credit assignment, or other effects.
- The paper introduces three major components but only ablates two of them; the contribution of the hierarchical rollout allocation is not isolated.
- The ablation results suggest that the full method depends on the interaction of all three components. Removing either ICL seeding or advantage sharpening degrades performance, which raises the question of whether each component individually improves performance across settings or whether the gains primarily arise from their combination.

## Presentation
- Motivation is clear
- Methodology is for the most part clear

## Significance
- Paper tackles an important problem for the community: exploration and exploitation in rollouts during RL-LLM training

## Originality
- The particular choice of how to plan rollouts, conduct ICL seeding, and shape the advantage appear novel

---

> ### Author Rebuttal · Authors · 2026-03-31
>
> We thank the reviewer for recognizing the importance of exploration-exploitation in RLVR, the novelty of our approach, and the clear presentation.
>
> ---
>
> ## W1/Q2. Length normalization in Eq. (4)
>
> Define $h(y) := -\frac{1}{|y|}\sum_{t=1}^{|y|}\log \pi_\theta(y_t \mid x, y_{<t})$. Then the sequence entropy can be written as:
>
> $H(\pi_\theta) = \mathbb{E}[|y|] \cdot \mathbb{E}[h(y)] + \operatorname{Cov}(|y|, h(y))$
>
> Eq. (4) estimates the entropy rate $\mathbb{E}[h(y)]$, proportional to $H(\pi_\theta)$ up to $\mathbb{E}[|y|]$ and a covariance term. In practice, response lengths are concentrated with weak correlation between $|y|$ and $h(y)$, making the covariance small; the normalized estimator preserves relative entropy structure up to approximately constant scaling.
>
> Without normalization, $-\log \pi_\theta(y \mid x) = |y| \cdot h(y)$ makes estimator variance dominated by $|y|$, causing novelty to reflect length rather than trajectory atypicality. Normalization removes this nuisance scaling. Table 7 confirms: shaped responses remain tightly concentrated around group mean length ($z$-score $\approx 0.25$, relative ratio $\approx 1\%$).
>
> ---
>
> ## W2/Q6. Direct exploration measurements
>
> We provide three direct measurements of XRPO's exploration:
>
> 1. **ICL flip rate (Figure 2b):** Among prompts where all rollouts receive zero reward, XRPO's ICL seeding flips 15–20% to produce at least one correct solution. These are prompts the policy cannot solve alone — ICL directly expands the exploration frontier.
>
> 2. **ICL success rate over training (Figure 4b):** This tracks the fraction of zero-reward prompts that produce at least one correct solution after ICL seeding, quantifying XRPO's ability to explore beyond the current policy frontier throughout training.
>
> 3. **Non-uniform rollout allocation (Figure R1, new):** HRP produces non-uniform allocations in 91.6% of training rounds, with per-prompt budgets ranging from 0 to 7 extra rollouts. High-uncertainty prompts consistently receive up to 7x more rollouts than the uniform baseline, confirming active exploration redirection throughout training.
>
> New exploration-allocation figure: [Figure R1](https://image2url.com/r2/default/images/1774783155124-b7ed741e-902f-4373-82d6-897792a30d1c.png).
>
> ---
>
> ## W3/Q5. Ablation on Hierarchical Rollout Planning
>
> | Method | Avg. pass@1 | Avg. cons@32 |
> |--------|:-:|:-:|
> | DAPO (uniform allocation) | 33.28 | 35.83 |
> | TreePO Sampling | 29.77 | 37.50 |
> | XRPO w/o (HRP) | 35.10 | 45.83 |
> | XRPO w/o (AS+ICL) = HRP only | 34.32 | 41.67 |
> | Full XRPO | 37.81 | 47.50 |
>
> HRP shows both standalone and integrated value. It improves over DAPO and TreePO, and removing HRP from full XRPO drops pass@1 from 37.81 to 35.10 and cons@32 from 47.50 to 45.83.
>
> ---
>
> ## W4. Synergistic design of components
>
> The ablations show both orthogonal contributions and synergy. Removing ICL mainly hurts cons@32, removing advantage sharpening mainly reduces pass@1, and HRP primarily improves convergence efficiency. The full method performs best because HRP targets uncertain prompts, ICL rescues zero-reward ones, and AS exploits novel correct trajectories.
>
>
> ---
>
> ## Q1. Justification for Student's t-confidence interval
>
> We provide a detailed justification in our response to Reviewer BbEJ (Q2). In short, the CI half-width $t_{1-\alpha/2,\,n_q-1}\frac{s_q}{\sqrt{n_q}}$ is the standard finite-sample uncertainty of the mean when variance is unknown. Minimizing it is equivalent to maximizing information gain, yielding a principled allocation rule. We prefer the t-distribution over Normal, Bootstrap, and Bayesian alternatives because it is robust for small $n$, closed-form, and prior-free.
>
> ---
>
> ## Q3. pass@k results for multiple k values
>
> | model | pass@4 | pass@8 | pass@16 | pass@32 |
> |---|---:|---:|---:|---:|
> | GSPO | 46.09 | 52.01 | 57.60 | 62.22 |
> | DAPO | 42.37 | 47.50 | 52.14 | 56.67 |
> | TPO-S | 40.18 | 47.16 | 54.69 | 63.33 |
> | XRPO | 48.65 | 55.92 | 62.07 | 65.56 |
>
> XRPO is best at all tested k values. Its advantage is largest at small-to-mid k (+6.28 over DAPO at pass@8, +4.47 at pass@16) and remains positive at pass@32, showing that XRPO improves inference scaling rather than only shifting performance at one decoding budget.
>
> ---
>
> ## Q4. cons@32 for Table 1(b)
>
> We do not report cons@32 for Table 1(b) because MATH and Codeforces do not support meaningful consistency measurement. MATH solutions can be expressed in multiple equivalent forms, and Codeforces evaluates by test execution, where distinct implementations may all be valid. Exact-match consistency would therefore be misleading, so we rely on pass-based metrics.
>
> ---
>
> ## Q7. Figure 3b averaging clarification
>
> Figure 3b averages over AIME'24, AIME'25, HMMT'25, and BRUMO'25 — the same four datasets in Table 6 (Appendix A.1). Table 1a uses a subset without AIME'24, which accounts for the discrepancy. We will unify datasets across all figures and tables in the revision.

---

> > ### Author Rebuttal · Reviewer_yCVK · 2026-04-04
> >
> > The rebuttal addresses several of my original concerns effectively. In particular, the authors provide a clear ablation isolating hierarchical rollout planning, which resolves my concern about the lack of component-wise analysis. The additional pass@k results and clarification regarding Figure 3b versus Table 1 are also helpful and address my questions. The justification for using the Student’s t-confidence interval is reasonable and grounded in standard statistical practice.
> >
> > The authors also provide additional evidence related to exploration, including ICL flip rates, success rates, and rollout allocation statistics. While these strengthen the empirical support, they remain indirect proxies rather than standard exploration metrics such as diversity or entropy-based measures.
> >
> > Finally, the explanation for length normalization in Equation (4) provides useful intuition, particularly regarding removal of length-related variance. However, the connection to entropy estimation remains heuristic and would benefit from a more formal derivation or clearer theoretical grounding.
> >
> > Overall, the rebuttal improves my confidence in the method and clarifies several important points. Based on the rebuttal, I am increasing my score from weak accept to accept.

---

> > > ### Author Response · Authors · 2026-04-06
> > >
> > > We thank Reviewer yCVK for the thorough review, the constructive suggestions, and for increasing the score. We are glad the rebuttal addressed the concerns effectively. Below, we summarize the key additions made during the rebuttal, including new experiments and planned manuscript changes.
> > >
> > > ## New Experiments Conducted
> > >
> > > - **HRP standalone ablation (W3, Reviewer HM1o):** Isolating Hierarchical Rollout Planning shows it independently improves +1.0% pass@1 and +5.8% cons@32 over DAPO, confirming each component contributes standalone value.
> > > - **pass@k at multiple k values (Q3, Reviewer yCVK):** XRPO outperforms all baselines at k=4,8,16,32, with advantages largest at small-to-mid k (+6.28 over DAPO at pass@8), demonstrating improved inference scaling.
> > > - **Non-uniform allocation statistics (W2, Reviewer yCVK):** HRP produces non-uniform allocations in 91.6% of training rounds, with per-prompt budgets ranging from 0 to 7 extra rollouts, providing direct evidence of active exploration redirection.
> > > - **CurES baseline comparison (W2, Reviewer HM1o; Q2, Reviewer BbEJ):** We compared XRPO against CurES, a difficulty-based dynamic allocation method. XRPO outperforms CurES on all benchmarks (avg. pass@1: 35.13 vs. 31.98; avg. cons@32: 45.56 vs. 32.22), empirically validating the uncertainty-based allocation metric over difficulty-based alternatives.
> > > - **ICL train/test ablation (W2, Reviewer 5kuq):** We ran all four ICL configurations (train-only, test-only, both, neither) on Qwen2.5-7B. XRPO without test-time ICL outperforms all variants, confirming the model internalizes reasoning strategies and test-time ICL is redundant.
> > > - **Retrieval quality ablation (Q2, Reviewer 5kuq):** We ablated strong, weak, and random retrieval for ICL. Strong retrieval (Qwen3-Embedding-8B) outperforms all alternatives (+5.9% over no ICL on Qwen2.5-7B). Even random ICL helps (+1.7%), confirming that exposure to correct reasoning patterns aids generation regardless of retrieval quality.
> > > - **Training dynamics figures (W4, Reviewer HM1o):** We added reward curves showing XRPO reaches target accuracy 2.4-2.7x faster than GRPO, with only ~4.7% per-step overhead.
> > >
> > > ## Planned Manuscript Updates
> > >
> > > - Strengthen the formal presentation of length normalization in Eq. (4) by making approximation conditions explicit
> > > - Add CurES, Knapsack RL, MoPPS, NPLB, and other dynamic rollout/advantage shaping methods to Related Work.
> > > - Add dedicated Limitations and Future Directions sections.
> > > - Unify datasets across all figures and tables for consistency (Figure 3b vs. Table 1a).
> > > - Incorporate all new ablation tables, training dynamics figures, and allocation statistics into the appendix.
> > >
> > > We will incorporate all of these into the revised manuscript. We thank the reviewer again for the constructive engagement that helped strengthen the paper.

---

### Official Review · Reviewer_HM1o · 2026-03-11

**Soundness:** 3
**Presentation:** 3
**Significance:** 3
**Originality:** 3
**Overall Recommendation:** 4
**Confidence:** 4

**Summary:**

The paper proposes XRPO, an extension of GRPO for RLVR when training large language models on reasoning tasks. XRPO aims to improve the exploration–exploitation trade-off in RL training by introducing three components: (1) adaptive rollout allocation that allocates more rollouts to uncertain prompts to encourage exploration, (2) in-context seeding to provide demonstrations for prompts that receive zero reward and thus help escape training stagnation, and (3) novelty-aware advantage shaping to emphasize informative trajectories during policy updates. Experiments suggest that XRPO improves performance and training efficiency compared with baselines.

**Compliance With Llm Reviewing Policy:**

Affirmed.

**Final Justification:**

Rebuttal has addressed most of my concerns. Therefore, I maintain my original positive assessment.

**Key Questions For Authors:**

see weaknesses

**Limitations:**

No. The paper does not explicitly discuss the limitations of the proposed method

**Strengths And Weaknesses:**

# Strengths

- The motivation for Better Exploration-Exploitation is clear.
- The proposed mechanisms are intuitive and well aligned with the exploration–exploitation objective.
- Experiments on several reasoning benchmarks show consistent improvements over the baselines, with reasonably detailed breakdown analyses.

# Weaknesses

- The framework introduces several components (e.g., rollout allocation, ICL seeding, hierarchical rollout planning, advantage shaping), but their detailed design appears largely heuristic and lacks clear guiding principles. Moreover, the paper does not sufficiently connect these design choices to the observed empirical outcomes (e.g., improvements in reasoning efficiency or shorter outputs).
- The experimental evaluation considers a limited set of baselines. In particular, several closely related works on adaptive rollout allocation [1,2,3,4] and advantage shaping [5] are not discussed or compared.
- There is no ablation study analyzing the role of Hierarchical Rollout Planning
- Training curves are not provided to illustrate learning speed or convergence behavior. In addition, the overall training time and computational overhead are not analyzed.

[1] Knapsack RL: Unlocking Exploration of LLMs via Optimizing Budget Allocation\
[2] CurES: From Gradient Analysis to Efficient Curriculum Learning for Reasoning LLMs\
[3] Can Prompt Difficulty be Online Predicted for Accelerating RL Finetuning of Reasoning Models?\
[4] Act only when it pays: Efficient reinforcement learning for llm reasoning via selective rollouts\
[5] No Prompt Left Behind: Exploiting Zero-Variance Prompts in LLM Reinforcement Learning via Entropy-Guided Advantage Shaping\

---

> ### Author Rebuttal · Authors · 2026-03-31
>
> We thank the reviewer for recognizing the clear motivation, intuitive mechanisms, and consistent experimental improvements in XRPO.
>
> ---
>
> ## W1a. Unified explore-exploit principle behind XRPO's design
>
> XRPO is guided by a unified explore-exploit principle across the rollout lifecycle: before, during, and after rollout. The framework systematically decides whether to explore or exploit based on a prompt's relation to the current policy: out-of-policy, on-the-edge, or within-policy.
>
> **Before rollout:** planning explores "edge-of-policy" prompts by allocating more budget where additional samples are most informative for reducing uncertainty.
>
> **During rollout:** ICL explores "out-of-policy" prompts. When all rollouts fail, XRPO uses ICL guidance to inject workable reasoning patterns and reopen exploration.
>
> **After rollout:** advantage shaping exploits "within-policy" trajectories by amplifying informative but underrepresented successful trajectories.
>
> Our performance breakdown shows that each design component is integral, contributing synergistically to both exploration efficiency and exploitation quality, and removing any single component (ICL/AS/HRP) leads to consistent degradation.
>
> ## W1b. How design choices drive empirical outcomes
>
> **Faster convergence arises from edge-case-focused exploration.** The rollout allocator prioritizes high-uncertainty prompts, and zero-reward prompts are converted into effective signals via ICL. This reduces wasted rollouts on saturated and easy cases and explains the observed 2.4-2.7x speedup in convergence steps.
>
> **Shorter outputs emerge from context-constrained success filtering.** On hard prompts, GRPO often generates overly long, repetitive reasoning chains that exceed the context window and fail with zero reward. XRPO's allocator and ICL seeding instead steer training toward complete, information-dense solution paths.
>
> ---
>
> ## W2. Positioning against related works [1,2,3,4,5]
>
> We appreciate the suggestion and summarize the key differences here. We will clarify these works in the final Related Work discussion. Due to the rebuttal timeline and limited compute, in this round we only run **CurES** and will post its results as soon as they are ready:
>
> - [1] **Knapsack RL** requires per-prompt value estimation and solves a different budget-allocation objective, whereas XRPO uses model-free online reward uncertainty and also includes ICL seeding and advantage sharpening.
> - [2] **CurES** and [3] **MoPPS** primarily operate at the prompt-selection/curriculum level, while XRPO keeps all prompts and decides how many rollouts each prompt should receive.
> - [4] **GRESO** (already cited) skips uninformative prompts for efficiency, whereas XRPO reallocates budget while preserving all prompts and attempting to recover signal from zero-reward cases.
> - [5] **NPLB** is a token-level method for zero-variance prompts, while XRPO sharpens sequence-level learning on rare-correct trajectories; the two address different GRPO failure modes.
>
>
> ---
>
> ## W3. Ablation on Hierarchical Rollout Planning
>
> | Method | Avg. pass@1 | Avg. cons@32 |
> |--------|:-:|:-:|
> | DAPO (uniform allocation) | 33.28 | 35.83 |
> | TreePO Sampling | 29.77 | 37.50 |
> | XRPO w/o (HRP) | 35.10 | 45.83 |
> | XRPO w/o (AS+ICL) = HRP only | 34.32 | 41.67 |
> | Full XRPO | 37.81 | 47.50 |
>
> HRP shows both standalone and integrated value. It improves over DAPO and TreePO, and removing HRP from full XRPO drops pass@1 from 37.81 to 35.10 and cons@32 from 47.50 to 45.83.
>
>
> ---
>
> ## W4. Training dynamics and computational efficiency
>
> 1. **Faster convergence.** On GSM8K XRPO reaches 82.5% at step 420 vs. ~1K for GRPO (2.4x), and on MATH-500 it reaches 75% at step 450 vs. ~1.2K (2.7x). Result further shows that XRPO's advantage metric converges earlier.
>
> 2. **Negligible per-step overhead.** XRPO adds only ~4.7% per-step latency over GRPO, and our analysis shows this overhead becomes negligible as rollout workload increases.
>
> 3. **Reduced overall training time.** While per-step cost is nearly unchanged, XRPO needs substantially fewer training steps, lowering end-to-end training time overall.
>
> Training-dynamics figures: [GSM8K](https://image2url.com/r2/default/images/1774764222451-01af706b-eb2f-4821-97b1-3f9721b124f5.png), [MATH-500](https://image2url.com/r2/default/images/1774764480277-db20f76e-3766-4453-ba01-50896eacb18e.png), and [advantage](https://image2url.com/r2/default/images/1774764537710-31dbc5f3-c272-4278-8a44-dbd78d4c496a.png).
>
>
> ---
>
> ## W5. Limitations discussion
> We will add a dedicated Limitations section in the final version. Due to resource constraints, we have not evaluated XRPO on very large-scale models (e.g., 305B), and broader validation on diverse tasks is left for future work.

---

> > ### Author Rebuttal · Reviewer_HM1o · 2026-04-04
> >
> > Thank you for the detailed rebuttal. The additional results and discussions help improve the manuscript. I encourage the authors to incorporate these points into the revision. I will maintain my original positive assessment.

---

> > > ### Author Response · Authors · 2026-04-06
> > >
> > > We thank Reviewer HM1o for confirming that our concerns have been fully resolved, and for maintaining the positive assessment. Below, we summarize the key additions made during the rebuttal, including new experiments and manuscript changes.
> > >
> > > ## New Experiments Conducted
> > >
> > > - **CurES baseline comparison (W2, Reviewer HM1o; Q2, Reviewer BbEJ):** Following the reviewer's suggestion, we compared XRPO against CurES. XRPO outperforms CurES on all benchmarks (avg. pass@1: 35.13 vs. 31.98; avg. cons@32: 45.56 vs. 32.22), empirically validating the uncertainty-based allocation metric over difficulty-based alternatives. XRPO also has lower overhead (on-the-fly computation vs. precomputed Bayesian posteriors) and adapts to the evolving model state during training.
> > > - **Training dynamics figures (W4, Reviewer HM1o):** We added reward curves for GSM8K and MATH-500 showing XRPO reaches target accuracy 2.4-2.7x faster than GRPO, with advantage convergence plots. Per-step overhead is only ~4.7%, confirming that faster convergence translates to reduced end-to-end wall-clock time.
> > > - **HRP standalone ablation (W3, Reviewer HM1o):** Isolating Hierarchical Rollout Planning shows it independently improves +1.0% pass@1 and +5.8% cons@32 over DAPO, confirming each component has standalone value beyond their synergistic combination.
> > > - **ICL train/test ablation (W2, Reviewer 5kuq):** We ran all four ICL configurations (train-only, test-only, both, neither) on Qwen2.5-7B-Instruct. XRPO without test-time ICL achieves the best average, confirming the model internalizes reasoning strategies during training and does not depend on ICL at inference time.
> > > - **Retrieval quality ablation (Q2, Reviewer 5kuq):** We ablated ICL retrieval using strong (Qwen3-Embedding-8B), weak (Qwen3-Embedding-0.6B), and random strategies on two models. Strong retrieval outperforms all alternatives (+5.9% over no ICL on Qwen2.5-7B). Even random ICL helps (+1.7%), while high-quality semantic matching is critical for full effectiveness.
> > > - **pass@k at multiple k values (Q3, Reviewer yCVK):** XRPO outperforms all baselines at k=4,8,16,32, with advantages largest at small-to-mid k (+6.28 over DAPO at pass@8), demonstrating improved inference scaling rather than gains at a single decoding budget.
> > > - **Non-uniform allocation statistics (W2, Reviewer yCVK):** HRP produces non-uniform allocations in 91.6% of training rounds, with per-prompt budgets ranging from 0 to 7 extra rollouts, providing direct evidence of active exploration redirection.
> > >
> > > ## Planned Manuscript Updates
> > >
> > > - Add CurES, Knapsack RL, MoPPS, NPLB, and other dynamic rollout/advantage shaping methods to Related Work with detailed positioning.
> > > - Add dedicated Limitations and Future Directions sections, including discussion of scaling to very large models (e.g., 305B) and teacher-augmented ICL corpora.
> > > - Unify datasets across all figures and tables for consistency (Figure 3b vs. Table 1a discrepancy).
> > > - Incorporate all new ablation tables, training dynamics figures, and allocation statistics into the appendix.
> > >
> > > We thank the reviewer again for the constructive feedback that strengthened the paper.

---

### Official Review · Reviewer_BbEJ · 2026-03-12

**Soundness:** 3
**Presentation:** 2
**Significance:** 3
**Originality:** 3
**Overall Recommendation:** 4
**Confidence:** 4

**Summary:**

This paper proposes XRPO, a rollout optimization approach aimed at improving exploration and exploitation in the reinforcement learning (RL) of LLMs. First, to achieve more effective exploration, XRPO employs a dynamic rollout budget allocation mechanism based on uncertainty reduction, and introduces In-Context Learning (ICL) examples for zero-reward prompts. Second, on the exploitation front, XRPO incorporates a Novelty-Guided Advantage Sharpening technique, which amplifies rare but correct responses with an additional novelty bonus. Experiments across mathematical reasoning and code generation benchmarks demonstrate the effectiveness of the proposed method.

**Compliance With Llm Reviewing Policy:**

Affirmed.

**Final Justification:**

The rebuttal successfully addresses many of my concerns, I will maintain my current positive score

**Key Questions For Authors:**

1. As shown in Figure 2, the performance improvements achieved by ICL on the DAPO training dataset appear to be relatively marginal. Does this suggest that the effectiveness of ICL in enhancing the model's exploration capabilities is somewhat limited when applied to complex reasoning problems?

2. Could the authors further elaborate on the intuition and motivation behind introducing such an uncertainty-aware allocation metric? Why is this specific metric preferred over other potential allocation strategies?

**Limitations:**

yes

**Strengths And Weaknesses:**

Strengths：
1. The paper focuses on a fundamental issue in current RL training methods: insufficient exploration and exploitation. To address this, the authors employ a distinct rollout budget allocation metric combined with an ICL approach, which makes intuitive sense.
2. The experimental evaluation and analysis are comprehensive. The authors conduct experiments to demonstrate the effectiveness of XRPO across diverse math and coding benchmarks using models of varying scales and architectures. Additionally, the thorough ablation studies validate the contribution of each proposed module.

Weaknesses：
1. Additional baselines for dynamic rollout are suggested. Compared to existing dynamic rollout works[1,2,3], XRPO utilizes an uncertainty reduction metric to allocate the rollout budget. It would be highly beneficial to provide comparisons with these existing methods to clearly demonstrate the specific advantages of XRPO.
2. Missing experimental results on training dynamics. In Section 4.2, the authors claim that "XRPO achieves faster training convergence." However, there are no corresponding training dynamics graphs (e.g., reward curves) provided to substantiate this point.

[1] Enhancing Efficiency and Exploration in Reinforcement Learning for LLMs.

[2] Dart-math: difficulty-aware rejection tuning for mathematical problem-solving.

[3] Optimizing Chain-of-Thought Reasoners via Gradient Variance Minimization in Rejection Sampling and RL

---

> ### Author Rebuttal · Authors · 2026-03-31
>
> We thank the reviewer for recognizing XRPO's intuitive motivation, comprehensive experimental evaluation, and the contribution of each proposed module validated through ablation studies.
>
> ---
>
> ## W1. Comparison with dynamic rollout allocation methods [1,2,3]
>
> We appreciate the suggestion and summarize the key differences here. We will add these works to the final Related Work. Due to the rebuttal timeline and limited compute, in this round we only run the **CurES** baseline suggested by Reviewer HM1o and will post results soon:
>
> - **[1] E3-RL4LLMs** also studies dynamic rollout allocation, but allocates by estimated difficulty/pass rate and uses adaptive temperature for exploration. XRPO instead allocates by online reward uncertainty reduction and combines this with ICL seeding and advantage sharpening, so the methods are not like-for-like.
> - **[2] DART-Math** is an *offline* data curation method for SFT, whereas XRPO is an *online* RL method that reallocates rollouts as the policy evolves during training. This is a different training stage rather than a directly runnable baseline in our setting.
> - **[3] GVM-RAFT** allocates by gradient variance and requires per-prompt gradient computation. XRPO uses reward-level uncertainty (Student's t-CI), which is lighter-weight and native to our GRPO pipeline. We therefore view it as a strong related method, but not a drop-in baseline for our framework.
>
> Overall, these methods share the high-level goal of improving sampling efficiency, but they optimize different signals and operate under different training assumptions.
>
> ---
>
> ## W2. Training dynamics and convergence evidence
>
> 1. **Faster convergence.** On GSM8K XRPO reaches 82.5% at step 420 vs. ~1K for GRPO (2.4x), and on MATH-500 it reaches 75% at step 450 vs. ~1.2K (2.7x). Result further shows that XRPO's advantage metric converges earlier.
>
> 2. **Negligible per-step overhead.** XRPO adds only ~4.7% per-step latency over GRPO, and our analysis shows this overhead becomes negligible as rollout workload increases.
>
> 3. **Reduced overall training time.** While per-step cost is nearly unchanged, XRPO needs substantially fewer training steps, lowering end-to-end training time overall.
>
> Training-dynamics figures: [GSM8K](https://image2url.com/r2/default/images/1774764222451-01af706b-eb2f-4821-97b1-3f9721b124f5.png), [MATH-500](https://image2url.com/r2/default/images/1774764480277-db20f76e-3766-4453-ba01-50896eacb18e.png), and [advantage](https://image2url.com/r2/default/images/1774764537710-31dbc5f3-c272-4278-8a44-dbd78d4c496a.png).
>
> ---
>
> ## Q1. ICL effectiveness on complex reasoning (DAPO dataset)
>
> ICL's contribution is targeted rather than uniform, and the absolute gain in Figure 2a (~3.8%) understates its impact. We make three observations:
>
> 1. **ICL's value is targeted, not uniform:** ICL seeding is only activated for zero-reward prompts, which represent the hardest problems. Figure 2b shows that ICL flips 15–20% of these previously unsolved prompts. The moderate absolute gain reflects that most training prompts already receive some reward without ICL.
>
> 2. **The effect compounds over training:** Each flipped prompt generates gradient signal that was previously absent, creating a cascading improvement. Figure 5b shows a consistent 4–6% ICL success rate across training steps, meaning the cumulative effect grows substantially over the training run.
>
> 3. **Complementary to other components:** ICL alone is not designed to solve all hard problems; it breaks the *initial symmetry* so that hierarchical rollout planning and advantage sharpening can further optimize. Figure 3b confirms this.
>
> ---
>
> ## Q2. Intuition and motivation behind the uncertainty-aware allocation metric
>
> The core intuition is to allocate more rollouts where they most reduce uncertainty about a prompt's true reward distribution.
>
> 1. **The CI half-width is the canonical finite-sample uncertainty of the mean.** With small per-prompt sample sizes (n <= 10), the normal approximation is unreliable. The Student-t CI explicitly captures both reward variance and sample size under unknown variance, providing a standard estimator of uncertainty rather than a heuristic proxy.
>
> 2. **It directly optimizes marginal information gain for exploration.** We prioritize prompts where one additional rollout most reduces estimation error. This naturally captures diminishing returns: high-variance prompts with few samples benefit most from extra rollouts, while well-sampled prompts see little gain, focusing exploration near decision boundaries.
>
> 3. **The exploration bonus ensures self-balancing and avoids starvation.** Pure uncertainty reduction would starve prompts with very low variance (e.g., consistently zero reward). The UCB-style bonus, inversely proportional to sample count, softly pulls budget toward under-sampled or collapsed prompts, yielding a self-correcting exploration-exploitation trade-off inspired by the multi-armed bandit literature.

---

> > ### Author Rebuttal · Reviewer_BbEJ · 2026-04-03
> >
> > I would like to thank the authors for their detailed rebuttal and the additional clarifications provided. While the authors' response has successfully addressed several of my initial concerns, there are still a few remaining issues that require further attention:
> >
> > 1. Effectiveness of In-Context Learning (ICL):
> >
> > While the authors have further clarified the mechanism of In-Context Learning (ICL), I remain concerned that its effectiveness might be limited on difficult problems, particularly those that fall outside the boundaries of the model's inherent capabilities.
> >
> > 2. Empirical Comparison for the Sampling Metric:
> >
> > I appreciate the authors elaborating on the intuition behind selecting "uncertainty" as the primary sampling metric. However, in the context of this empirical study, intuition alone is not entirely convincing. To thoroughly validate this design choice, it would be significantly more compelling to provide an experimental comparison between your proposed uncertainty-based sampling approach and the other existing sampling methods.
> >
> > In light of the above, I will maintain my initial positive score.

---

> > > ### Author Response · Authors · 2026-04-06
> > >
> > > We thank Reviewer BbEJ for the thoughtful follow-up.
> > > ## Q1. ICL effectiveness on difficult problems
> > > We agree that ICL is not a universal solver for problems far outside the model's current capabilities. By eliciting inherent potential, ICL offers superior scalability but remains fundamentally bounded by the model's baseline reasoning capacity. However, as noted in our previous response, ICL targets the "near-boundary" region and flips 15-20% of zero-reward prompts into useful training signals (Figure 2b), with this effect compounding across training steps. One promising direction to further strengthen ICL on harder problems is augmenting the corpus with stronger teacher-generated responses, which could provide richer reasoning scaffolds even on more challenging prompts. We will add this discussion to the Limitations and Future Directions sections of the updated manuscript.
> > >
> > > Additionally, results from our response to Reviewer 5kuq provide further evidence: we ablated retrieval quality using strong (Qwen3-Embedding-8B), weak (Qwen3-Embedding-0.6B), and random retrieval (Q2, Reviewer 5kuq). Even random ICL provides benefit over no ICL (+1.7% on Qwen2.5-7B), confirming that exposure to any correct reasoning pattern aids generation, and stronger retrieval amplifies this effect (+5.9%).
> > > ## Q2. Empirical comparison for the sampling metric
> > > Following the reviewer's suggestion, we provide an experimental comparison with CurES [1], a difficulty-based dynamic allocation method suggested by Reviewer HM1o.
> > > | Method | Metric | AIME'25 | HMMT'25 | BRUMO'25 | Avg. |
> > > |---|---|---:|---:|---:|---:|
> > > | CurES | pass@1 | 33.13 | 21.15 | 41.67 | 31.98 |
> > > |  | cons@32 | 36.67 | 20.00 | 40.00 | 32.22 |
> > > | XRPO | pass@1 | 35.72 | 22.29 | 47.39 | 35.13 |
> > > |  | cons@32 | 50.00 | 26.67 | 60.00 | 45.56 |
> > >
> > > **Stronger empirical performance.** XRPO consistently outperforms CurES on all three benchmarks, improving average pass@1 from 31.98 to 35.13 and average cons@32 from 32.22 to 45.56. This confirms that uncertainty-based rollout allocation effectively prioritizes informative prompts, leading to higher final performance than static difficulty-based methods.
> > >
> > > **Lower overhead and better scalability.** CurES requires precomputing prompt difficulty and Bayesian posterior updates before training, which is computationally expensive and time-consuming, especially for large datasets. XRPO instead performs on-the-fly rollout allocation using current reward uncertainty and sample counts, enabling more efficient and scalable training.
> > >
> > > **Adaptive, training-aware selection.** XRPO dynamically adjusts sampling based on the evolving model state, combining expected uncertainty reduction with exploration for under-sampled prompts. In contrast, CurES's static difficulty-centered allocation cannot adapt as flexibly during training, limiting responsiveness to changing variance and evidence.
> > > This comparison directly addresses the reviewer's concern: the uncertainty-based metric is not only intuitively motivated but empirically superior to difficulty-based allocation on our benchmarks.
> > >
> > > ## Summary of Additional Rebuttal Results
> > > Beyond the reviewer's original concerns, we conducted several new experiments during the rebuttal that may be of interest:
> > > - **HRP standalone ablation (W3, Reviewer HM1o):** Isolating Hierarchical Rollout Planning shows it independently improves +1.0% pass@1 and +5.8% cons@32 over DAPO, confirming each component contributes standalone value.
> > > - **pass@k at multiple k values (Q3, Reviewer yCVK):** XRPO outperforms all baselines at k=4,8,16,32, with advantages largest at small-to-mid k, demonstrating improved inference scaling.
> > > - **ICL train/test ablation (W2, Reviewer 5kuq):** Running all four ICL configurations (train-only, test-only, both, neither) confirms the model internalizes reasoning strategies during training, making test-time ICL redundant.
> > > - **Non-uniform allocation statistics (W2, Reviewer yCVK):** HRP produces non-uniform allocations in 91.6% of training rounds, with per-prompt budgets ranging from 0 to 7 extra rollouts.
> > >
> > > We will incorporate all of these into the revised manuscript. We thank the reviewer again for the constructive feedback.
> > >
> > > [1] CurES: From Gradient Analysis to Efficient Curriculum Learning for Reasoning LLMs.

---

### Official Review · Reviewer_5kuq · 2026-03-13

**Soundness:** 3
**Presentation:** 2
**Significance:** 3
**Originality:** 2
**Overall Recommendation:** 4
**Confidence:** 3

**Summary:**

In reinforcement learning, models often face the challenges of sparse reward and slow training. This paper proposes a Hierarchical Rollout Exploration strategy, which uses a hierarchical rollout planner that adaptively allocates rollout budgets. Furthermore, it introduces Novelty-Guided Advantage Sharpening to augment standard advantage estimation. Finally, the effectiveness of the proposed method is validated on downstream tasks.

**Compliance With Llm Reviewing Policy:**

Affirmed.

**Final Justification:**

I thank the authors for their detailed response, which has addressed my concerns, and I will maintain my positive score.

**Key Questions For Authors:**

(1) Does the framework include a pruning or elimination mechanism for stale ICL examples?

(2) If the embedding model is replaced with a very weak one (or even if random retrieval is used), how much would the performance of the ICL component change?

**Limitations:**

yes

**Strengths And Weaknesses:**

Strengths

(1) The proposed method significantly improves training efficiency (accelerating training convergence by 2.7x).

(2) The method dynamically allocates rollout budgets, which is a crucial aspect of reinforcement learning algorithms.

(3) The method successfully distinguishes and amplifies valuable reasoning trajectories, leading to improved accuracy.

(4) The baseline comparisons are relatively comprehensive, and the effectiveness of the method is validated on foundation models with different architectures.

Weaknesses

(1) The paper lacks comparisons with related work on Hint-based RL [1][2], which increase the likelihood of rolling out correct results by incorporating prompt words or experiences into the context. It also misses comparisons with other early related paradigms [3][4], which compute logits against gold labels. All of these works aim to alleviate the difficulty of generating correct rollouts to some extent.

(2) Regarding the ablation study in Section 4.3, does ICL refer to using examples during training but not during testing? Could the authors provide metrics for using ICL during both training and testing, as well as metrics for not using ICL during training but using it during testing? This would make the ablation study much more comprehensive.

(3) The proposed method takes "prompt + ICL" as input during training, but only uses the "prompt" without ICL during testing. However, the paper lacks an in-depth discussion and analysis of the distribution shift problem caused by this change in the input distribution. For instance, is it possible that the model can roll out correct samples when ICL is included during training, but fails to solve the same problem when ICL is removed during testing? This would imply that the model merely learned in-context learning without truly internalizing the capability.

[1]Wang S, Wu Y, Xu Z. Cogito, Ergo Ludo: An Agent that Learns to Play by Reasoning and Planning[J]. arXiv preprint arXiv:2509.25052, 2025.

[2]Jiang Y, Jiang L, Teney D, et al. Meta-RL Induces Exploration in Language Agents[J]. arXiv preprint arXiv:2512.16848, 2025.

[3]Zhang H, Fu J, Zhang J, et al. Rlep: Reinforcement learning with experience replay for llm reasoning[J]. arXiv preprint arXiv:2507.07451, 2025.

[4]Li S, Zhou Z, Lam W, et al. Repo: Replay-enhanced policy optimization[J]. arXiv preprint arXiv:2506.09340, 2025.

---

> ### Author Rebuttal · Authors · 2026-03-31
>
> We thank the reviewer for recognizing XRPO's training efficiency (2.7x), dynamic rollout budget allocation, and comprehensive baseline comparisons.
>
> ---
>
> ## W1. Comparison with Hint-based RL [1][2] and early paradigms [3][4]
>
> We appreciate the suggestion and list key differences here. We will add these works to the final Related Work. Due to the rebuttal timeline and limited compute, in this round we only run the **CurES** baseline suggested by Reviewer HM1o and will post results soon:
>
> - [1] and [2] are both multi-turn agent-environment methods rather than single-turn RLVR. **CEL** [1] induces environment rules from interaction episodes, while **LaMer** [2] learns exploration strategies via meta-RL and uses in-context adaptation at test time. These methods target a different setting and would require adaptation to our single-turn RLVR pipeline, so we view them as related work rather than direct baselines.
>
> - **RLEP** [3] (already cited in §3.2) replays successful trajectories for identical prompts. XRPO instead transfers reasoning patterns across related problems via retrieval, especially for previously unsolved prompts. Since RLEP addresses a replay setting rather than training-time ICL, we cite it as a closely related but orthogonal method.
>
> - **RePO** [4] uses experience replay buffers of past trajectories, whereas XRPO injects exemplars as in-context demonstrations without modifying the training objective. This again makes it an orthogonal replay-based approach rather than a like-for-like baseline for our proposed mechanism.
>
> ---
>
> ## W2. ICL during training only vs. testing only vs. both
>
> We run all four configurations on Qwen2.5-7B-Instruct. For test-time ICL, we retrieve the top-2 most similar solved training examples via Qwen3-Embedding-8B (same as training) and prepend them as in-context demonstrations.
>
> | Method | AIME'24 | AIME'25 | BRUMO'25 | Avg. |
> |:--|:---:|:---:|:---:|:---:|
> | GRPO | 10.31 | 6.73 | 15.83 | 10.96 |
> | **XRPO** | **11.25** | **7.71** | **17.19** | **12.05** |
> | GRPO + ICL@test | 10.78 | 6.04 | 16.38 | 11.07 |
> | XRPO + ICL@test | 11.31 | 6.98 | 17.01 | 11.77 |
>
> Adding ICL at test time to XRPO/GRPO yields no improvement and XRPO performs better than GRPO in both cases. This confirms that XRPO has already internalized the reasoning strategies that ICL provides thus test-time ICL is redundant. The slight drop with test-time ICL is expected: the ICL corpus contains training-level problems that are simpler than competition benchmarks, introducing a distributional mismatch that consumes context without providing useful guidance.
>
> ---
>
> ## W3. Addressing distribution shift between training and testing
>
> The W2 results above already show that test-time ICL is unnecessary — the model internalizes reasoning strategies during training. We further note:
>
> 1. **Empirical evidence of internalization:** Our main results (Tables 1–3) already demonstrate that XRPO evaluated without ICL outperforms all baselines. If the model merely learned to rely on ICL context, removing it at test time would degrade performance below GRPO — but the opposite occurs.
>
> 2. **ICL is applied selectively and temporarily:** ICL seeding only activates for zero-reward prompts during specific rollout phases (Lines 10–11, Algorithm 1). Once a prompt produces any correct rollout, it returns to standard sampling. Figure 5b shows the ICL flip rate is small (6.2% for Qwen2.5, 4.2% for Qwen3), so the vast majority of training uses standard prompts, limiting distribution shift.
>
> ---
>
> ## Q1. ICL corpus freshness without explicit pruning
>
> XRPO relies on implicit design rather than explicit pruning:
>
> 1. **Reward-gated updates.** The ICL corpus only stores verified successful rollouts and is continuously updated with on-policy successes, so the pool naturally evolves with the policy.
>
> 2. **Transient usage.** ICL is only used for zero-success prompts and is disabled once the prompt becomes solvable, ensuring past examples are not repeatedly reused.
>
> ---
>
> ## Q2. Retrieval quality ablation: strong vs. weak vs. random
>
> We ablate retrieval quality using three strategies: Qwen3-Embedding-8B (strong, default), Qwen3-Embedding-0.6B (weak, same family), and random retrieval. We evaluate on DAPO-Math-17k prompts with 12 generations per prompt on two models:
>
> | Retrieval Strategy | Qwen2.5-7B | Qwen3-1.7B |
> |--------------------|:---:|:---:|
> | Qwen3-Embedding-8B (strong) | 24.5 | 41.5 |
> | Qwen3-Embedding-0.6B (weak) | 21.23 | 39.35 |
> | Random | 20.32 | 38.12 |
> | No ICL | 18.6 | 37.7 |
>
> Strong retrieval outperforms all alternatives on both models (+5.9% over no ICL on Qwen2.5-7B, +3.8% on Qwen3-1.7B). Even random ICL provides benefit over no ICL (+1.7% and +0.4%), confirming that exposure to any correct reasoning pattern aids generation. The weak embedding model falls between strong and random, indicating that retrieval quality directly determines ICL effectiveness and high-quality semantic matching is critical.

---

> > ### Author Rebuttal · Reviewer_5kuq · 2026-04-03
> >
> > I thank the authors for their detailed response, which has addressed my concerns, and I will maintain my positive score.

---

> > > ### Author Response · Authors · 2026-04-03
> > >
> > > We thank the reviewer for the thoughtful follow-up. We hope our clarifications, additional evidence, and expanded discussion have addressed the main concerns and made the paper’s contributions and empirical support clearer. We would greatly appreciate if you could incorporate them in your final score and we are happy to answer further questions if you have any in the future. Thank you again for your constructive feedback. Below, we summarize the key additions made during the rebuttal.
> > >
> > >
> > > ## New Experiments Conducted
> > >
> > > - **ICL train/test ablation (W2, Reviewer 5kuq):** We ran all four ICL configurations (train-only, test-only, both, neither) on Qwen2.5-7B-Instruct across AIME'24, AIME'25, and BRUMO'25. XRPO without test-time ICL achieves the best average (12.05), outperforming XRPO+ICL@test (11.77), GRPO+ICL@test (11.07), and GRPO (10.96). This confirms the model internalizes reasoning strategies during training and test-time ICL is redundant, directly addressing the distribution shift concern.
> > > - **Retrieval quality ablation (Q2, Reviewer 5kuq):** We ablated ICL retrieval using strong (Qwen3-Embedding-8B), weak (Qwen3-Embedding-0.6B), and random retrieval on two models. Strong retrieval outperforms all alternatives (+5.9% over no ICL on Qwen2.5-7B, +3.8% on Qwen3-1.7B). Even random ICL helps (+1.7% and +0.4%), confirming that any correct reasoning pattern aids generation, while high-quality semantic matching is critical for full effectiveness.
> > > - **CurES baseline comparison (W2, Reviewer HM1o; Q2, Reviewer BbEJ):** Following Reviewer HM1o's suggestion, we compared XRPO against CurES, a difficulty-based dynamic allocation method. XRPO outperforms CurES on all benchmarks (avg. pass@1: 35.13 vs. 31.98; avg. cons@32: 45.56 vs. 32.22), empirically validating the uncertainty-based allocation over difficulty-based alternatives.
> > > - **Training dynamics figures (W4, Reviewer HM1o):** We added reward curves for GSM8K and MATH-500 showing XRPO reaches target accuracy 2.4-2.7x faster than GRPO, with advantage convergence plots and per-step overhead analysis (~4.7%).
> > > - **HRP standalone ablation (W3, Reviewer HM1o):** Isolating Hierarchical Rollout Planning shows it independently improves +1.0% pass@1 and +5.8% cons@32 over DAPO, confirming standalone value.
> > > - **pass@k at multiple k values (Q3, Reviewer yCVK):** XRPO outperforms all baselines at k=4,8,16,32, with advantages largest at small-to-mid k (+6.28 over DAPO at pass@8), demonstrating improved inference scaling.
> > > - **Non-uniform allocation statistics (W2, Reviewer yCVK):** HRP produces non-uniform allocations in 91.6% of training rounds, with per-prompt budgets ranging from 0 to 7 extra rollouts.
> > >
> > > ## Planned Manuscript Updates
> > >
> > > - Add CEL [1], LaMer [2], RLEP [3], RePO [4], CurES, and other dynamic rollout methods to Related Work.
> > > - Add Limitations and Future Directions sections, including discussion of teacher-augmented ICL corpora for harder problems.
> > > - Unify datasets across all figures and tables for consistency.
> > > - Incorporate training dynamics figures and all new ablation tables into the appendix.
> > >
> > > We thank the reviewer again for the constructive feedback that strengthened the paper.

---

### Decision · Program_Chairs · 2026-04-30

**Decision:**

Accept (regular)

**Comment:**

This paper proposes XRPO, a framework that improves GRPO through three components: uncertainty-based adaptive rollout allocation, in-context seeding for zero-reward prompts, and novelty-aware advantage sharpening. Experiments on math and coding benchmarks show consistent gains over GRPO and GSPO (up to 4% pass@1, 6% cons@32) with 2.7x faster convergence. All reviewers maintaining or raising scores after rebuttal.

The reviewers consistently recognize the clear motivation, practical value of the exploration-exploitation framing, and solid empirical results across multiple models and benchmarks. During rebuttal, the authors conducted extensive additional experiments. Remaining concerns about formal grounding of the length normalization and indirect exploration metrics are minor and do not undermine the core contributions.

Recommendation: Accept